# CROSS-DOMAIN FEW-SHOT CLASSIFICATION VIA INVARIANT-CONTENT FEATURE RECONSTRUCTION

## ABSTRACT

In *cross-domain few-shot classification* (CFC), mainstream studies aim to fast train a new module to select or transform features (a.k.a., the high-level semantic features) for previously unseen domains with a few labeled training data available on top of a powerful pre-trained model. These studies usually *assume* that high-level semantic features are shared across these domains, and just simple feature selection or transformations are enough to adapt features to those unseen domains. However, in this paper, we find that the simply transformed features are too general to fully cover the key content features regarding each class. Thus, we propose *invariant-content feature reconstruction* (IFR) to train a simple module that simultaneously consider high-level and fine-grained invariant-content features for the previously unseen domains. Specifically, the fine-grained invariant-content features are considered as a set of *informative* and *discriminative* features learned from a few labeled training data of tasks sampled from unseen domains, and are extracted by retrieving features that are invariant to style modifications from a set of content-preserving augmented data in pixel level with an attention module. Extensive experiments on the Meta-Dataset benchmark show that IFR achieves good generalization performance on unseen domains, which demonstrates the effectiveness of the fusion of the high-level features and the fine-grained invariant-content features. Specifically, IFR improves the average accuracy on unseen domains by 1.6% and 6.5% respectively under two different CFC experimental settings.

## 1 INTRODUCTION

Deep learning has revealed strong ability to deal with various learning tasks (e.g. the classification task) since convolutional neural network (CNN) (LeCun et al., 1989) was proposed. Although great progress has been achieved, there still exist two problems that hinder the wide application of deep learning techniques. Firstly, a model that is able to perform well on test data requires to be trained on a large amount of labeled training data which are often expensive to obtain and sometimes unavailable. Moreover, directly applying a pre-trained model to data sampled from an unseen domain usually fails to achieve satisfactory generalization performance because of the distribution discrepancies between the source and target domains (Chi et al., 2021; Kuzborskij & Orabona, 2013).

A feasible learning paradigm for solving the aforementioned problems is *cross-domain few-shot classification* (CFC) which learns to perform classification on tasks sampled from unseen domains with only a few labeled training samples available. Due to the scarce data and distribution gaps, mainstream works (Dvornik et al., 2020; Liu et al., 2021a; Li et al., 2021) propose to train a simple module, such as a linear head, on top of a single or multiple pre-trained feature extractors to transform or select features for target domains. These studies *implicitly assume* that there exist common high-level semantic representations that are shared across datasets (Dvornik et al., 2020; Liu et al., 2021a). To be specific, a pre-trained model is able to recognize and extract previously observed high-level semantic features from data sampled from unseen domains. For example, a model that is pre-trained on ImageNet (Russakovsky et al., 2015) is expected to perform well on VGG Flower (Nilsback & Zisserman, 2008) and CU Birds (Wah et al., 2011), since high-level semantic patterns of flowers and birds have been observed in ImageNet dataset during the pre-training phase.

However, we find that simply transformed features which mainstream studies merely focus on are too general to fully cover the key content features of the target class. Take URL (Li et al., 2021) as an

example, as shown in Fig. 1, the features learned with URL (Li et al., 2021), a representative state-of-the-art method in cross-domain few-shot classification that trains a simple linear transformation head to map the universal representations extracted from a pre-trained multi-domain backbone to the task-specific space, fails to capture the comprehensive and representative semantic features for the target classes. Such phenomenon motivates us to think which kind of features ought to be learned from labeled data for cross-domain few-shot classification tasks sampled from unseen domains.

In this paper, we take two aspects into consideration. On the one side, the learned features should be *informative* enough to contain comprehensive semantic information of the target class. On the other hand, the learned features are expected to be *discriminative*. In other words, the learned features should be representative for the target class and robust to trivial information changes. An intuitive illustration of the aforementioned expected features is shown in Fig. 1. We can recognize the guitar in Fig. 1 as a combination of nuts, tuning pegs and frets even if the trivial style information (e.g. the color) of the image changes. In this example, the idea of considering the guitar as a combination of nut, pegs and frets provides a set of informative features of guitar in the image. Besides, the unchanged features like tuning pegs represent the discriminative information that are robust to trivial information (e.g., style information) changes.

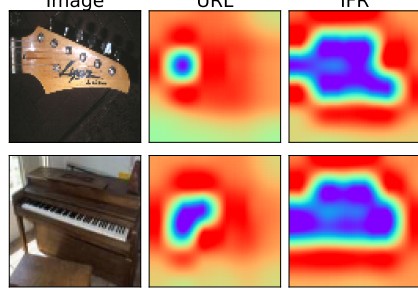

Figure 1: Visualizations of original images and corresponding features learned respectively with URL and our proposed IFR methods. Compared with URL, IFR tends to capture more comprehensive and representative semantic features.

In practice, we formulate the aforementioned desirable features as a kind of *fine-grained invariant-content features*, and further propose an effective approach called *invariant-content feature reconstruction* (IFR) to consider both high-level representations and fine-grained invariant-content features simultaneously. An inspiration of IFR is that pixels, which contain invariant-content information in an original image, ought to be highly similar to their corresponding parts in the content-preserving augmented counterpart. To be specific, we adopt content-preserving augmentation techniques, such as cropping, flipping, color jittering and grayscaling, to mimic the case that the style information of an image is modified. Then, the similarity between the original image and its augmented counterpart is measured in pixel level to determine the invariant-content features. Finally, the invariant-content features are reconstructed by retrieving feature pixels from the augmented data according to the similarity matrix obtained above. The complete pipeline of IFR method is illustrated in Fig. 2

The performance of our IFR approach has been extensively evaluated on Meta-Dataset (Triantafillou et al., 2020). The experimental results manifest that IFR can achieve good generalization performance on previously unseen domains, which demonstrates the effectiveness of the fusion of high-level representations and fine-grained invariant-content features. To be concrete, IFR improves the average accuracy on unseen domains by 1.6% and 6.5% respectively under two different settings.

## 2 RELATED WORK

**Cross-Domain Few-Shot Classification.** Cross-domain few-shot classification aims to learn to perform classification on previously unseen data and domains with only few labeled training data available. Existing works usually pose such problem in a *learning-to-learn* (Schmidhuber, 1987; Bengio et al., 1990; Thrun & Pratt, 2012) paradigm, which is also know as meta-learning Finn et al. (2017); Snell et al. (2017). Generally, current cross-domain few-shot classification approaches can be mainly divided into two different branches. The first one is training a pipeline from scratch (Triantafillou et al., 2020; Baik et al., 2020; Requeima et al., 2019; Bateni et al., 2020).

Different from aforementioned methods, other works tend to leverage the powerful pre-trained backbone. SUR Dvornik et al. (2020) proposes to learn a coefficient vector to linearly combine the representations extracted from several independent pre-trained domain-specific backbones. URT Liu et al. (2021a) proposes a multi-head Universal Representation Transformer Vaswani et al. (2017) layer to select feature maps. Due to the several forward passes of all backbones, computational cost of SUR and URT during inference time is quite expensive. Thus, URL Li et al. (2021) proposes to

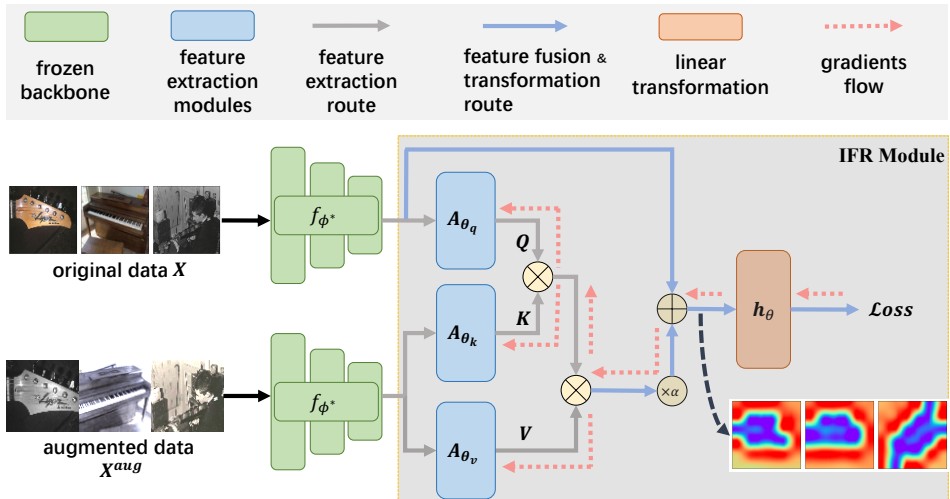

Figure 2: **Illustration of IFR pipeline.** The IFR method considers both high-level representations and fine-grained invariant-content features simultaneously with a meticulously designed module. IFR first extracts the fine-grained invariant-content features with a single attention head (consists of a query head $A_{\theta_q}$, a key head $A_{\theta_k}$ and a value head $A_{\theta_v}$) by retrieving content features that are invariant to style modifications (simulated by content-preserving data augmentations), then the high-level representations and invariant-content features are fused and transformed by a linear transformation $h_\theta$ for further classification. For comparison, the illustration of URL pipeline is provided in Fig. 7.

distill a single powerful backbone from the domain-specific backbones and simply train a classifier on top of the frozen pre-trained backbone with few labeled samples of the new task during inference.

**Feature Reconstruction.** Feature reconstruction is widely applied in many previous works Baker & Matthews (2004); Cao et al. (2014); Sun et al. (2015), and recently attracts attention from few-shot classification for learning specific representations. DeepEMD Zhang et al. (2020) decomposes an image into a set of semantic representations and reconstructs the target image from the perspective of solving an optimal transport problem. FRN Wertheimer et al. (2021) formulates the linear reconstruction of query features as an ridge regression problem with a closed-form solution. CrossAttention Hou et al. (2019) and CrossTransformer Doersch et al. (2020) project the query features into the space of support data and makes predictions by measuring the distances of class-conditioned projections and targets. For more detailed introduction about related works, please see Appendix A.

## 3  INVARIANT-CONTENT FEATURE RECONSTRUCTION

In this section, we first briefly review the problem setup for cross-domain few-shot classification and then introduce the core idea of reconstructing invariant-content features in details. Finally, we describe our IFR pipeline which takes both high-level and fine-grained invariant-content features into consideration simultaneously when performing cross-domain few-shot classification.

### 3.1  PROBLEM SETUP FOR CFC

Let $\mathcal{S} = \{\mathcal{S}_i\}_{i=1}^{|\mathcal{S}|}$ denote the meta-dataset that contains several subdatasets. Each subdataset owns 3 disjoint sets: $\mathcal{S}_i = \{\mathcal{D}_{\mathcal{S}_i}^{tr}, \mathcal{D}_{\mathcal{S}_i}^{val}, \mathcal{D}_{\mathcal{S}_i}^{test}\}$. We denote the subdatasets in which the training sets are accessible to training as $\mathcal{S}^{seen}$ while the remaining as $\mathcal{S}^{unseen}$, i.e., $\mathcal{S} = \mathcal{S}^{seen} \cup \mathcal{S}^{unseen}$, $\mathcal{S}^{seen} \cap \mathcal{S}^{unseen} = \emptyset$. Thus, the training set $\mathcal{D}^{tr}$, validation set $\mathcal{D}^{val}$ and test set $\mathcal{D}^{test}$ under cross-domain few-shot classification setting can be formulated as:

$$\mathcal{D}^{tr} = \mathcal{D}_{\mathcal{S}_1}^{tr} \cup \mathcal{D}_{\mathcal{S}_2}^{tr} \cup ... \cup \mathcal{D}_{\mathcal{S}_{|\mathcal{S}^{seen}|}}^{tr},$$

$$\mathcal{D}^{val} = \mathcal{D}_{\mathcal{S}_1}^{val} \cup \mathcal{D}_{\mathcal{S}_2}^{val} \cup ... \cup \mathcal{D}_{\mathcal{S}_{|\mathcal{S}^{seen}|}}^{val},$$

$$\mathcal{D}^{test} = \mathcal{S} \backslash (\mathcal{D}^{tr} \cup \mathcal{D}^{val}).$$

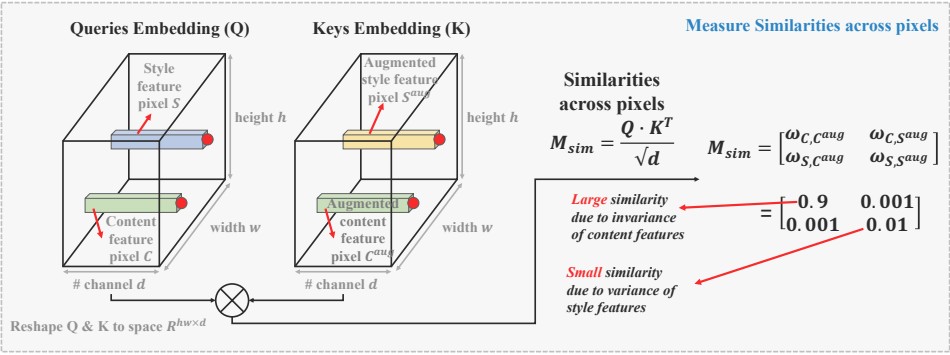

(a) Similarity measurement

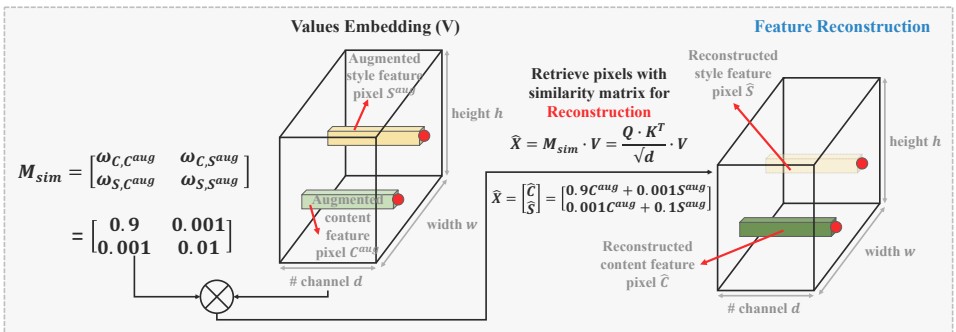

(b) Feature reconstruction

Figure 3: **Illustration of Invariant-content Feature Reconstruction. (a) Similarity Measurement.** Given embeddings of queries (original images) and keys (augmented images) $Q, K \in \mathbb{R}^{w \times h \times d}$, a similarity matrix $M_{\mathrm{sim}} \in \mathbb{R}^{wh \times wh}$ is generated for the similarities among samples. Since we assume that content features (green) are invariant to style (blue and yellow) modifications (e.g. colors etc.), larger scores will be obtained from the comparison between two context feature pixels. **(b) Feature Reconstruction.** The content feature reconstruction is equivalent to linearly recombining the pixel vectors with corresponding similarity scores (weights). Pixel vectors assigned with large similarity scores are highlighted while those with small similarity scores are weakened.

For each learning episode in meta-training, meta-validation and meta-test phase, a task $\mathcal{T} = \{\mathcal{D}_\mathcal{T}, \mathcal{Q}_\mathcal{T}\}$ is sampled from corresponding dataset ($\mathcal{D}^{\mathrm{tr}}$ or $\mathcal{D}^{\mathrm{val}}$ or $\mathcal{D}^{\mathrm{test}}$), where $\mathcal{D}_\mathcal{T} = \{X^{\mathrm{s}}, Y^{\mathrm{s}}\} = \{(\boldsymbol{x}_i^{\mathrm{s}}, y_i^{\mathrm{s}})\}_{i=1}^{|\mathcal{D}_\mathcal{T}|}$ denotes support data pairs which are training data pairs of the task and $\mathcal{Q}_\mathcal{T} = \{X^{\mathrm{q}}, Y^{\mathrm{q}}\} = \{(\boldsymbol{x}_j^{\mathrm{q}}, y_j^{\mathrm{q}})\}_{j=1}^{|\mathcal{Q}_\mathcal{T}|}$ denotes query data pairs which are test data and labels of the task.

## 3.2 INVARIANT-CONTENT FEATURE RECONSTRUCTION

To begin with, we would like to first introduce two assumptions proposed in ReLIC (Mitrovic et al., 2020): (i) the two fundamental elements of an image are *content* and *style*, (ii) content and style are independent, i.e. style changes are content-preserving. According to these assumptions, content features that depict the discriminative information of an image are invariant to style modifications.

Motivated by the fact that representations learned by existing CFC approach are too general to fully cover the key content of the target class (see Fig. 1) and inspired by the aforementioned assumptions, we propose to leverage the invariant-content features in a fine-grained way to learn a set of informative features that are discriminative and robust to style information changes to perform cross-domain few-shot classification tasks. Such fine-grained invariant-content features mainly include two desirable merits. First of all, they are fine-grained and thus contain comprehensive and informative semantic information of the target class. Besides, they represent the most fundamental and discriminative features of the target class and are robust to trivial style information (e.g., the color of the image) changes. The inspiration behind invariant-content feature reconstruction is that the pixels that contain invariant-content features in original images should highly resemble the corresponding parts of their augmented counterparts since content features do not vary with style

changes. In order to mimic the case that style information is modified while the content information is preserved, we follow ReLIC and adopt cropping, flipping, color jittering and grayscaling as the *content-preserving* augmentation techniques and randomly apply them to a set of support data.

Generally, the reconstruction of invariant-content features mainly include two steps: locating the positions of invariant content features and retrieving these features. In this work, we adopt a simple attention head (Vaswani et al., 2017) to perform invariant-content feature reconstruction (see Fig. 3).

Specifically, we first locate the positions of invariant-content features by measuring the similarities between original and augmented image representations in pixel level. As illustrated in Fig. 3(a), given a set of queries embedding and keys embeddings, $\boldsymbol{Q} \in \mathbb{R}^{w \times h \times d}$ (transformed from original images) and $\boldsymbol{K} \in \mathbb{R}^{w \times h \times d}$ (transformed from augmented images), a similarity matrix $\boldsymbol{M}_{\mathrm{sim}} \in \mathbb{R}^{wh \times wh}$ is measured to describe the similarity between arbitrary two pixels respectively from queries and keys embedding. As the aforementioned assumptions, content features are invulnerable to the changes of style. Thus, two pixels that depict the same content information ought to be highly similar to each other and the similarity score will be relatively larger than those depict trivial style information. Thus, the positions of

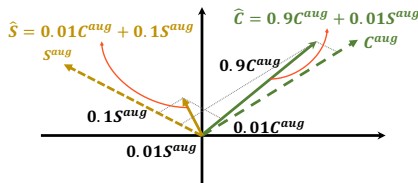

Figure 4: **Illustration for effect of multiplication in feature reconstruction.** Large weights help highlight the corresponding vectors while small weights weaken the corresponding vectors.

pixels that depict the same invariant-content features in augmented representations are determined for each pixel of original representations. Then, with the similarity matrix, the invariant-content features are reconstructed by retrieving pixels with corresponding similarity scores (as shown in Fig. 3(b)). The reconstruction process is equivalent to a recombination of pixel vectors where the invariant-content features assigned with larger similarity scores are highlighted (as shown in Fig. 4).

### 3.3 IFR PIPELINE

In this section, we describe the complete learning pipeline of our proposed invariant-content feature reconstruction method in details. A brief illustration of IFR pipeline is available in Fig. 2.

**Augmented Data Generation.** At the beginning of each episode, a batch of augmented data is generated with the original support data by randomly imposing a series of content-preserving data augmentations. To be specific, given a randomly sampled task $\mathcal{T} = \{\mathcal{D}_{\mathcal{T}}, \mathcal{Q}_{\mathcal{T}}\}$, where $\mathcal{D}_{\mathcal{T}} = \{X^{\mathrm{s}}, Y^{\mathrm{s}}\} = \{(\boldsymbol{x}_i^{\mathrm{s}}, y_i^{\mathrm{s}})\}_{i=1}^{|\mathcal{D}_{\mathcal{T}}|}$ and $\mathcal{Q}_{\mathcal{T}} = \{X^{\mathrm{q}}, Y^{\mathrm{q}}\} = \{(\boldsymbol{x}_j^{\mathrm{q}}, y_j^{\mathrm{q}})\}_{j=1}^{|\mathcal{Q}_{\mathcal{T}}|}$, we can correspondingly obtain an augmented task $\mathcal{T}^{\mathrm{aug}} = \{\mathcal{D}_{\mathcal{T}}, \mathcal{D}_{\mathcal{T}}^{\mathrm{aug}}, \mathcal{Q}_{\mathcal{T}}\}$, where $\mathcal{D}_{\mathcal{T}}^{\mathrm{aug}} = \{X^{\mathrm{a}}, Y^{\mathrm{a}}\} = \{(\boldsymbol{x}_u^{\mathrm{a}}, y_u^{\mathrm{a}})\}_{u=1}^{b|\mathcal{D}_{\mathcal{T}}|}$ and $b$ is an integer number, by randomly applying content-preserving data augmentations to $X^{\mathrm{s}}$. In particular, for each data point in original support data, $b$ augmented samples can be generated.

**Invariant-content Feature Reconstruction.** Given an aforementioned augmented task $\mathcal{T}^{\mathrm{aug}} = \{\mathcal{D}_{\mathcal{T}}, \mathcal{D}_{\mathcal{T}}^{\mathrm{aug}}, \mathcal{Q}_{\mathcal{T}}\}$. Let $f_{\phi^*}(\boldsymbol{x}) \in \mathbb{R}^{c \times h \times w}, \boldsymbol{x} \in \mathcal{T}^{\mathrm{aug}}$ denote the output of the pre-trained backbone, where $c$ is the number of channel, $h$ and $w$ are the height and width of the output. We use $A_{\theta_{\mathrm{q}}}, A_{\theta_{\mathrm{k}}}$ and $A_{\theta_{\mathrm{v}}}$ to respectively denote the queries, keys and values heads of a single attention head. The subscript $\theta$ means the parameters of the head. Each head is a mapping: $A_\theta : \mathbb{R}^{c \times h \times w} \mapsto \mathbb{R}^{c \times h \times w}$.

Firstly, queries embedding $\boldsymbol{Q} \in \mathbb{R}^{|\mathcal{D}_{\mathcal{T}}| \times c \times h \times w}$, keys embedding $\boldsymbol{K} \in \mathbb{R}^{b|\mathcal{D}_{\mathcal{T}}| \times c \times h \times w}$ and values embedding $\boldsymbol{V} \in \mathbb{R}^{b|\mathcal{D}_{\mathcal{T}}| \times c \times h \times w}$ are obtained by applying the pre-trained backbone and the corresponding linear transformation: $A_\theta \circ f_{\phi^*}(\cdot)$. In our IFR, we treat both original support and query data as queries while the augmented support data as keys and values. For simplicity, we then flatten queries as $\bar{\boldsymbol{Q}} \in \mathbb{R}^{|\mathcal{D}_{\mathcal{T}}|hw \times c}$, keys as $\bar{\boldsymbol{K}} \in \mathbb{R}^{b|\mathcal{D}_{\mathcal{T}}|hw \times c}$ and values $\bar{\boldsymbol{V}} \in \mathbb{R}^{b|\mathcal{D}_{\mathcal{T}}|hw \times c}$ respectively.

Then, for each queries data point $\boldsymbol{q} \in \bar{\boldsymbol{Q}}, \boldsymbol{q} \in \mathbb{R}^{hw \times c}$, we follow Fig. 3 to measure the similarities across pixels and reconstruct the invariant-content features:

$$\hat{\boldsymbol{x}} = \boldsymbol{M}_{\mathrm{sim}} \cdot \bar{\boldsymbol{V}} = \mathrm{softmax}\left(\frac{\boldsymbol{q} \cdot \bar{\boldsymbol{K}}^\top}{\sqrt{c}}\right) \cdot \bar{\boldsymbol{V}}, \tag{1}$$

where $\hat{\boldsymbol{x}}$ denotes the reconstructed representations of the original image embedding $f_{\phi^*}(\boldsymbol{x})$, $\boldsymbol{M}_{\mathrm{sim}}$ denotes the similarity matrix of queries and keys, and $\mathrm{softmax}(\cdot)$ denotes the softmax function.

**Adaptation Strategy.** In this work, the proposed IFR module is trained in the same way as previous pre-trained-based cross-domain few-shot classification methods (Snell et al., 2017; Liu et al., 2021a; Li et al., 2021). Specifically, we first fuse the fine-grained reconstructed support data representations $\hat{X}^{\mathrm{s}}$ with the original high-level representations $f_{\phi^*}(X^{\mathrm{s}})$ and perform a linear transformation on the fused representations:

$$Z^{\mathrm{s}} = h_\theta\Big(f_{\phi^*}(X^{\mathrm{s}}) + \alpha \cdot \hat{X}^{\mathrm{s}}\Big), \tag{2}$$

where $Z^{\mathrm{s}}$ is the transformed fused features, and $\alpha$ is a scale coefficient of reconstructed fine-grained invariant-content representations. Then, the prototype $c_i$ for class $i$ can be reformualted as:

$$c_i = \frac{1}{|\mathcal{C}_i|} \sum_{z^{\mathrm{s}} \in \mathcal{C}_i} z^{\mathrm{s}}, \ \mathcal{C}_i = \{z^{\mathrm{s}}_j | y^{\mathrm{s}}_j = i\}, \tag{3}$$

where $z^{\mathrm{s}}_j \in Z^{\mathrm{s}}$, $y^{\mathrm{s}}_j$ is the label of $z^{\mathrm{s}}_j$ and $i \in \{1, 2, ..., N_c\}$. Then, the likelihood of $z^{\mathrm{s}}$ belonging to class $t$ is formulated as:

$$p(\hat{y} = t | z^{\mathrm{s}}) = \frac{e^{\cos(z^{\mathrm{s}}, c_t)}}{\sum_{l=1}^{N_c} e^{\cos(z^{\mathrm{s}}, c_l)}}. \tag{4}$$

Since IFR aims at learning a set of optimal parameters to maximize the likehood of samples belonging to their own classes, the objective loss of IFR is formulated as:

$$\min_{\{\theta_{\mathrm{q}}, \theta_{\mathrm{k}}, \theta_{\mathrm{v}}, \theta\}} -\frac{1}{|Z^{\mathrm{s}}|} \sum_{j=1}^{|Z^{\mathrm{s}}|} \log\Big(p(\hat{y} = y^{\mathrm{s}}_j | z^{\mathrm{s}}_j)\Big). \tag{5}$$

During meta-test phase, query data are treated as queries and reconstructed with Eq. (1). Then, the predictions are made with Eq. (5) following Eq. (4). (See Algorithm 1 for brief summary of IFR.)

### 3.4 ANALYSIS

In URL (Li et al., 2021), a simple linear transformation head is trained on top of a frozen pre-trained backbone. The merit of the linear transformation lies in that the linear transformation function space is of less complexity and thus robust, especially when the data in each few-shot classification task is scarce and the number of model parameters is small. However, the less-complex function space also limits the ability of model to represent complex functions. For example, when few-shot classification tasks are diverse, the less-complex function space might not be enough to complete all the tasks.

In contrast, due to its complicated compositions of several computational operators, the attention transformation function is of more complexity and, in turn, is able to represent more functions compared with the linear transformation function. Since the goal of IFR is to retrieve invariant-content features by measuring the similarities between arbitrary two pixels in original data and their augmented counterparts, we select the attention for feature reconstruction. However, such a complex transformation also raises a concern: whether the distance between two original samples that share similar content features becomes extremely large (e.g., infinity) after applying attention. To analyze this concern, we first introduce a strict definition of attention transformation (Bahdanau et al., 2014).

**Definition 1 (Attention (Bahdanau et al., 2014))** *Let $K = (k_1, ..., k_N) \subset \mathbb{R}^{d_k}$ be a collection of keys, $V = (v_1, ..., v_N) \subset \mathbb{R}^{d_v}$ a collection of corresponding values, and $q \in \mathbb{R}^{d_q}$ a query. Also, let $a : \mathbb{R}^{d_q} \times \mathbb{R}^{d_k} \to \mathbb{R}$ be a similarity function. Then attention is the mapping:*

$$\mathrm{Attention}(q, K, V) := \sum_{i=1}^{N} \mathrm{softmatch}_a(q, K)_i \cdot v_i,$$

*where $\mathrm{softmatch}_a(q, K)$ is a probability distribution over the elements of $K$ defined as:*

$$\mathrm{softmatch}_a(q, K)_i := \frac{\exp(a(q, k_i))}{\sum_{j=1}^{N} \exp(a(q, k_j))}.$$

Based on the above definition, recent work (Vuckovic et al., 2021) shows that the attention transformation function is Lipschitz continuous under mild assumptions.

Table 1: **Results on Meta-Dataset (Trained on all datasets).** Mean accuracy and 95% confidence are reported in the following table.

| Datasets | Proto-MAML | CNAPS | SimpleCNAPS | SUR | URT | FLUTE | Tri-M | 2LM | URL | **IFR(Ours)** |
|---|---|---|---|---|---|---|---|---|---|---|
| ImageNet | 46.5± 1.1 | 50.8±1.1 | 58.4 ±1.1 | 56.2 ± 1.0 | 56.8 ± 1.1 | **58.6 ± 1.1** | 51.8±1.1 | 58.0±3.6 | 57.2 ± 1.1 | 56.9 ± 1.1 |
| Omniglot | 82.7± 1.0 | 91.7±0.5 | 91.6 ± 0.6 | 94.1 ± 0.4 | 94.2 ± 0.4 | 92.0 ± 0.5 | 93.2±0.5 | **95.3±1.0** | 94.3 ± 0.4 | 94.6 ± 0.4 |
| Aircraft | 75.2± 0.8 | 83.7±0.6 | 82.0 ± 0.7 | 85.5 ± 0.5 | 85.8 ± 0.5 | 82.8 ± 0.5 | 87.2±0.5 | **88.2±0.5** | 88.1 ± 0.5 | 88.1 ± 0.5 |
| Birds | 69.9± 1.0 | 73.6±0.9 | 74.8 ± 0.9 | 71.0 ± 1.0 | 76.2 ± 0.8 | 75.3 ± 0.8 | 79.2±0.8 | **81.8±0.6** | 80.2± 0.8 | 80.1 ± 0.8 |
| Textures | 68.2± 1.0 | 59.5±0.7 | 68.8 ± 0.9 | 71.0 ± 0.8 | 71.6 ± 0.7 | 71.2 ± 0.8 | 68.8±0.8 | 76.3 ±2.4 | 76.3 ± 0.7 | **76.5 ±0.7** |
| Quick Draw | 66.8± 0.9 | 74.7±0.8 | 76.5 ±0.8 | 81.8 ± 0.6 | 82.4 ± 0.6 | **82.4 ± 0.6** | 77.3 ± 0.7 | 79.5±0.7 | 78.3±0.7 | 82.1 ± 0.6 | **82.4 ± 0.6** |
| Fungi | 42.0±1.2 | 50.2±1.1 | 46.6 ± 1.0 | 64.3 ± 0.9 | 64.0 ± 1.0 | 48.5 ± 1.1 | 58.1±1.1 | **69.6±1.5** | 68.0 ± 1.0 | 66.9 ± 1.0 |
| VGG Flower | 88.7± 0.7 | 88.9±0.5 | 90.5 ± 0.5 | 82.9 ± 0.8 | 87.9 ± 0.6 | 90.5 ± 0.6 | 91.6±0.6 | 90.3±0.8 | 91.9 ± 0.5 | **92.7 ± 0.5** |
| Traffic Sign | 52.4 ± 1.1 | 56.5 ±1.1 | 57.2 ± 1.0 | 51.0 ± 1.1 | 48.2 ± 1.1 | 63.0 ± 1.1 | 58.4±1.1 | 63.6±1.5 | 63.1 ± 1.1 | **66.8 ± 1.1** |
| MSCOCO | 41.7 ± 1.1 | 39.4 ±1.0 | 48.9 ± 1.1 | 52.0 ± 1.1 | 51.5 ± 1.1 | 52.8 ± 1.0 | 50.0±1.0 | **57.0±1.1** | 54.2 ± 1.0 | 55.9 ±1.0 |
| MNIST | - | - | 94.6 ± 0.4 | 94.3 ± 0.4 | 90.6 ± 0.5 | **96.2 ± 0.5** | 95.6±0.5 | 94.7±0.5 | 94.6 ± 0.4 | 95.2 ±0.4 |
| CIFAR-10 | - | - | 74.9 ± 0.7 | 66.5 ± 0.9 | 67.0 ± 0.8 | 75.4 ± 0.7 | **78.6±0.7** | 71.5±0.9 | 71.5 ± 0.8 | 72.6 ±0.8 |
| CIFAR-100 | - | - | 61.3 ± 1.1 | 56.9 ± 1.1 | 57.3 ± 1.0 | 62.0 ± 1.0 | **67.1±1.0** | 60.0±1.1 | 63.0 ± 1.0 | 64.0 ±1.0 |
| Average Seen | 67.5 | 71.6 | 73.7 | 75.9 | 77.4 | 74.5 | 76.2 | 79.7 | **79.8** | **79.8** |
| Average Unseen | - | - | 67.4 | 64.1 | 62.9 | 69.9 | 69.9 | 69.4 | 69.3 | **70.9** |
| Average All | - | - | 71.2 | 71.3 | 71.8 | 72.7 | 73.8 | 75.7 | 75.7 | **76.4** |
| Average Rank | - | - | 6.4 | 6.8 | 6.0 | 4.7 | 4.6 | 3.1 | 3.2 | **2.4** |

[1] Results of URL are the average of 5 random seeds. The reproduction results are consistent with the results reported on their website.

[2] The results of our method are the average of ten random reproduction experiments.

**Theorem 2 (Vuckovic et al. (2021))** *Let $K = \{k_1, ..., k_N\} \subset \mathbb{R}^d$ and $V = \{v_1, ..., v_N\} \subset \mathbb{R}^d$ and $\text{Attention}(\cdot, K, V)$ be the attention introduced in definition 1. Given the same mild assumptions in Theorem 14 (Vuckovic et al., 2021), then the mapping $q \mapsto \text{Attention}(q, K, V)$ is Lipschitz continuous:*

$$\|\text{Attention}(q_1, K, V) - \text{Attention}(q_2, K, V)\|_2 \leq L\|q_1 - q_2\|_2,$$

*where $L$ is a uniform constant.*

Theorem 2 shows that the distance between the features of two samples obtained from the attention module is upper bounded by the distance between the two samples with a uniform Lipschitz constant. Thus, given a well-pre-trained backbone, if the features obtained by the backbone are close, then, after the attention transformation, these features will still be close to each other, rather than that they are extremely far away. Based on this continuous property of the attention transformation function, we can reliably leverage IFR to find good representations for each few-shot classification task, although IFR is more complex than the linear transformation head used in URL.

## 4 EXPERIMENTS

In this section, we conduct extensive experiments to validate the performance of our IFR method. We first evaluate IFR on *vary-way vary-shot* tasks as proposed in the mainstream benchmark Meta-Dataset (Triantafillou et al., 2020) under two different experimental settings. Then, we perform a series of analysis and ablation studies to further explore the properties of components in IFR.

**Implementation Details.** Following previous works (Triantafillou et al., 2020; Dvornik et al., 2020; Liu et al., 2021a; Li et al., 2021), we use choose ResNet-18 (He et al., 2016) as the backbone and evaluate our method on Meta-Dataset (Triantafillou et al., 2020) (see Appendix C for details). For convenience and fairness, we directly use the ResNet-18 (He et al., 2016) backbones provided in URL repository[1]. Same as Li et al. (2021), we train a task-specific IFR module for each task sampled from unseen data and domains on top of the frozen pre-trained backbone. An IFR module is composed of a single attention head which includes three linear layers, and a linear transformation block which consists of a BN layer, an average pooling layer and a linear layer. At the beginning of each adaptation step, we re-initialize all linear layers in the IFR module with the identity matrix as done in URL (Li et al., 2021) for a better start to optimize the parameters by taking advantages of the learned features. Besides, the weights and biases of BN layer are re-initialized with 0 and 1 respectively. For query, key and value layers, a scale $1e - 4$ is multiplied to their weights. The optimizer used in our method is Adadelta (Zeiler, 2012). For more details, please refer to D.

---

[1]https://github.com/VICO-UoE/URL

Table 2: **Results on Meta-Dataset (Trained on ImageNet only)** Mean accuracy and 95% confidence are reported in the table.

| Datasets | Finetune | ProtoNets | ProtoNets (large) | BOHB | fo-Proto-MAML | ALFA+fo-Proto-MAML | FLUTE | URL | **IFR(Ours)** |
|---|---|---|---|---|---|---|---|---|---|
| ImageNet | 45.8±1.1 | 50.5±1.1 | 53.7±1.1 | 51.9±1.1 | 49.5±1.1 | 52.8±1.1 | 46.9±1.1 | 55.8±1.0 | **56.6±1.1** |
| Omniglot | 60.9±1.6 | 60.0±1.4 | 68.5±1.3 | 67.6±1.2 | 63.4±1.3 | 61.9±1.5 | 61.6±1.4 | 67.4±1.4 | **72.2±1.2** |
| Aircraft | **68.7±1.3** | 53.1±1.0 | 58.0±1.0 | 54.1±0.9 | 56.0±1.0 | 63.4±1.1 | 48.5±1.0 | 49.5±0.9 | 62.1±1.0 |
| Birds | 57.3±1.3 | 68.8±1.0 | **74.1±0.9** | 70.7±0.9 | 68.7±1.0 | 69.8±1.1 | 47.9±1.0 | 71.2±0.9 | **73.0±0.9** |
| Textures | 69.0±0.9 | 66.6±0.8 | 68.8±0.8 | 68.3±0.8 | 66.5±0.8 | 70.8±0.9 | 63.8±0.8 | 73.0±0.6 | **75.8±0.7** |
| Quick Draw | 42.6±1.2 | 49.0±1.1 | 53.3±1.0 | 50.3±1.0 | 51.5±1.0 | 59.2±1.2 | 57.5±1.0 | 53.9±1.0 | **61.6±1.0** |
| Fungi | 38.2±1.0 | 39.7±1.1 | 40.7±1.2 | 41.4±1.1 | 40.0±1.1 | 41.5±1.2 | 31.8±1.0 | 41.6±1.0 | **46.5±1.1** |
| VGG Flower | 85.5±0.7 | 85.3±0.8 | 87.0±0.7 | 87.3±0.6 | 87.2±0.7 | 86.0±0.8 | 80.1±0.9 | 87.0±0.6 | **88.7±0.6** |
| Traffic Sign | **66.8±1.3** | 47.1±1.1 | 58.1±1.1 | 51.8±1.0 | 48.8±1.1 | 60.8±1.3 | 46.5±1.1 | 47.4±1.1 | 64.6±1.1 |
| MSCOCO | 34.9±1.0 | 41.0±1.1 | 41.7±1.1 | 48.0±1.0 | 43.7±1.1 | 48.1±1.1 | 41.4±1.0 | 53.5±1.0 | **55.5±1.0** |
| MNIST | - | - | - | - | - | - | 80.8±0.8 | 78.1±0.7 | **89.1±0.7** |
| CIFAR-10 | - | - | - | - | - | - | 65.4±0.8 | 67.3±0.8 | **72.2±0.8** |
| CIFAR-100 | - | - | - | - | - | - | 52.7±1.1 | 56.6±0.9 | **63.5±1.0** |
| Average Seen | 45.8 | 50.5 | 53.7 | 51.9 | 49.5 | 52.8 | 46.9 | 55.8 | **56.6** |
| Average Unseen | - | - | - | - | - | - | 56.5 | 62.2 | **68.7** |
| Average All | - | - | - | - | - | - | 55.8 | 61.7 | **67.1** |
| Average Rank | 6.1 | 7.3 | 3.8 | 4.3 | 5.8 | 3.9 | 8.4 | 3.8 | **1.4** |

[1] The results of our method are the average of ten random reproduction experiments.

[2] The ranks only consider the first 10 datasets.

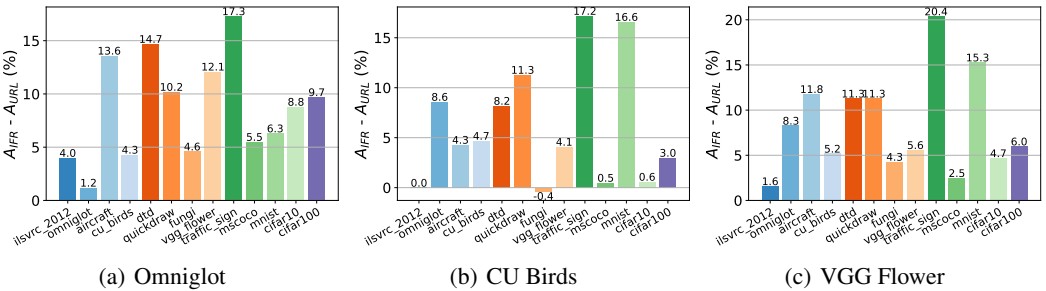

(a) Omniglot       (b) CU Birds       (c) VGG Flower

Figure 5: **Comparisons of evaluation results of IFR and URL with other domain-specific backbones.** We additionally evaluate IFR on other single domain-specific backbones and compare the test accuracies with URL. The results show that IFR consistently outperforms.

## 4.1 RESULTS ON META-DATASET

We evaluate our invariant-feature reconstruction method under two experimental settings: *'Training on all datasets'* and *'Training on ImageNet only'* Triantafillou et al. (2020). 'Training on all datasets' requires the pre-trained backbone to be exposed to several subdatasets during training phase while 'Training on ImageNet only' only requires the backbone to be pre-trained on ImageNet dataset (see Appendix C for more details). We compare our method with the several existing state-of-the-art methods, such as Proto-MAML (Triantafillou et al., 2020), ProtoNets (large) (Doersch et al., 2020), fo-Proto-MAML(FP-MAML) (Triantafillou et al., 2020), CNAPS (Requeima et al., 2019), SimpleC-NAPS (Bateni et al., 2020), BOHB (Saikia et al., 2020), Tri-M (Liu et al., 2021b), SUR (Dvornik et al., 2020), URT (Liu et al., 2021a), ALFA+fo-Proto-MAML(ALFA+FP-MAML) (Baik et al., 2020), FLUTE (Triantafillou et al., 2021), 2LM (Qin et al., 2023), and URL (Li et al., 2021).

**Training on All Datasets.** The experimental results under 'Train on all datasets' are reported in Table 1. Among all methods, IFR achieves the best performance in average and ranks 2.4. The average performance of our proposed IFR is comparable with URL Li et al. (2021) on seen datasets while better than all other methods on unseen datasets. Compared with URL, which performs better than other methods except ours in average, IFR obtains better results on Omniglot (+0.2%), Textures (+0.2%), QuickDraw (+0.3%), VGG Flower (+0.8%), Traffic Sign (+3.7%), MSCOCO (+1.7%), MNIST (+0.6%), CIFAR-10 (+1.1%) and CIFAR-100 (+1.0%). The results show that our method can achieve better generalization performance on unseen domains, which is more challenging than seen domains. Besides, we also notice that IFR outperforms recent work, 2LM (Qin et al., 2023), on 7 out of 13 datasets, and achieves better average performance on both seen and unseen domains.

**Training on ImageNet Only.** As we can observe from Table 2, IFR outperforms on 10 out of 13 datasets and ranks 1.4 in average. Compared with the second best approach among all state-of-

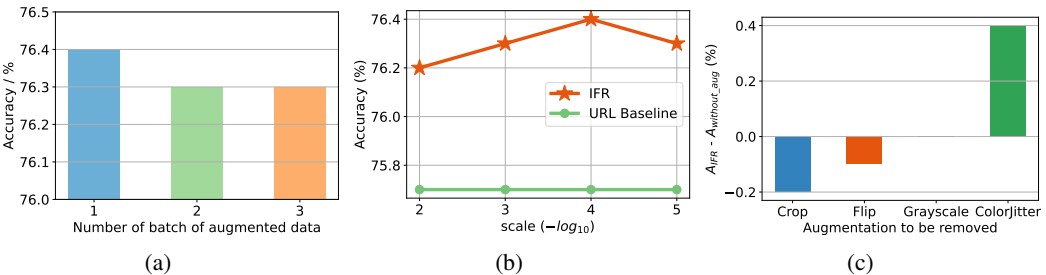

Figure 6: (a). Average performance on all datasets with different numbers of batch of the augmented data; (b). Comparison of scale coefficient ranging from $1e-2$ to $1e-5$; (c). Ablation study on augmentation techniques (ImageNet).

the-art methods, IFR performs better on ImageNet (+0.8%), Omniglot (+4.8%), Textures (+2.8%), QuickDraw (+2.4%), Fungi (+4.9%), VGG Flower (+1.5%), MSCOCO (+2.0%), MNIST (+8.3%), CIFAR-10 (+4.7%) and CIFAR-100 (+6.9%). Moreover, similar to the results under 'Train on All Datasets' settings, IFR consistently significantly outperforms URL on unseen domains.

Further, we also evaluate IFR with other single domain-specific backbones on Meta-Dataset. We reported part of the experimental results in Fig. 5. Complete results are available in Fig. 8 and Table 12. Fig. 5 depicts the differences between accuracies of IFR and URL. As shown in the figure, IFR consistently achieves much better generalization performance than URL on unseen domains.

## 4.2 FURTHER ANALYSES

**Number of Augmented Data.** An intuition of IFR is that more augmented data help extract better invariant-content features due to the abundant semantic features. However, the results in Fig. 6(a) show the opposite. The average total performance does not change obviously. However, according to the numerical results in Table 3, it is easy to observe that different datasets react differently to different numbers of augmented data. For example, Traffic Sign prefers more augmented data while MSCOCO, DTD prefer less. Moreover, datasets, such as Omniglot, MNIST and CIFAR, are not sensitive to the number of augmented data. From our perspective, a reasonable conjecture for such phenomenon is that a batch of augmented data has contain enough semantic features for simple datasets like MNIST while complex datasets such as MSCOCO require more semantic features.

**Scale Coefficient.** We further explore the effect of scale coefficient. As manifested in Fig. 6(b), IFR reaches its best performance (averaged accuracy on all datasets) when the scale coefficient $\alpha = 1e-4$. Generally, our IFR method is robust to $\alpha$ within the interval $[1e-5, 1e-2]$.

**Ablation Study for Augmentations.** We further conduct a series of ablation studies to explore the roles that the four augmentations play in the performance of all datasets (see Table 8). Generally, different augmentations have different effect on different datasets. Take ImageNet (Fig. 6(c)) as an example, ImageNet prefers color jittering to cropping, flipping and grayscaling. We conjecture the reason is that images in ImageNet contains too many objects and color jittering can help capture the key features while cropping may cut some important parts of the objects. Although removing some augmentations contributes to achieving better performance on a single dataset, better average performance on seen, unseen and all domains are achieved when applying all the four augmentations.

## 5 CONCLUSION

In this paper, we propose an effective invariant-content feature reconstruction method to consider both high-level and fine-grained invariant-content features simultaneously when performing cross-domain few-shot classification. We reconstruct the invariant-content features via retrieving content features, which contain informative and discriminative features of the images and are invariant to style modifications, from a set of augmented support data derived from original images. Our proposed IFR method aims at training a simple IFR module which is able to capture the invariant-content features and combine them with the high-level semantic features for cross-domain few-shot classification tasks. Extensive experiments on Meta-dataset benchmark, along with hyper-parameter analyses and ablation studies demonstrate that IFR can achieve good generalization performance on unseen domains and capture more comprehensive and representative features for target classes.

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

# A    DETAILED RELATED WORK

**Representations of Meta-Learning**: Many works have paid their efforts to few-shot classification problem by posing it in a *learning-to-learn* Schmidhuber (1987); Bengio et al. (1990); Thrun & Pratt (2012) paradigm, which is also known as meta-learning (Vinyals et al., 2016; Ravi & Larochelle, 2017; Finn et al., 2017; Snell et al., 2017). Among the existing methods, MAML (Finn et al., 2017) and Prototypical Networks (Snell et al., 2017) are the most widely applied learning frameworks. Prototypical Networks learns a powerful encoder that outputs a centroid for each class by averaging the features of support data so that the distances between query data of each class and the corresponding centroids is small. Different from Prototypical Nets, MAML learns a set of parameters which is treated as model initialization in each task episode from training tasks via solving a bi-level optimization problem so that it can be fast adapted to unseen tasks during inference phase. In order to improve the efficiency and the generalization performance, many variants of MAML emerge in recent works (Nichol et al., 2018; Lee et al., 2019; Bertinetto et al., 2019; Rusu et al., 2019; Tian et al., 2020a). Further, in order to figure out the essence of how MAML generalize to unseen tasks, Raghu et al. (2020) conducts a series of analyses on the representations learned with MAML and reveals that the impressive power of MAML derives from feature reuse instead of fast adaptation. Such conclusion inspires another type of few-shot classification approach. Different from those *end-to-end* few-shot learning methods that learn a model from scratch, the *pre-trained* few-shot learning pipelines propose to leverage the powerful pre-trained backbone, such as ResNet (He et al., 2016), to extract better representations and then fine-tune a classifier on top of the backbone during evaluation phase, such as Baseline++ (Chen et al., 2019; Tian et al., 2020b). Besides, Meta-Baseline (Chen et al., 2021) also proposes to fine-tune the entire model with a nearest-centroid similarity. Compared with the end-to-end approaches, the pre-trained pipelines are more flexible to fine-tune and usually achieve better generalization performance during inference phase since the frozen pre-trained embedding encoder can extract high-quality features and the classifier owns less parameters.

**Cross-Domain Few-Shot Classification:** Cross-domain few-shot classification is a challenging variant of typical few-shot classification task that aims to perform classification on unseen data with exposed to few labeled training samples. Different from typical few-shot classification task in which the data are sampled from the same distribution, cross-domain few-shot classification aims to achieve good generalization performance not only on the observed domains but also on unseen domains. Similar to meta-learning, cross-domain few-shot classification also contains 'training from scratch' and 'pre-training' pipelines. From the angle of training from scratch, due to the advantages

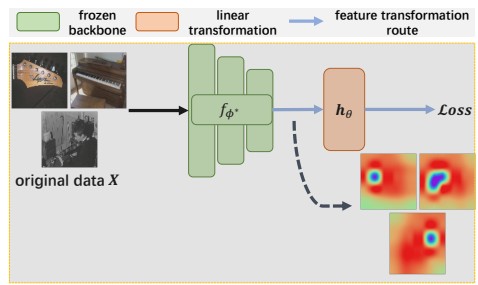

Figure 7: **The learning pipeline of URL.**

of Prototypical Nets (Snell et al., 2017) and MAML (Finn et al., 2017), Proto-MAML (Triantafillou et al., 2020) combines the two meta-learning paradigms by treating the prototypes learned from the embedding network as the parameters of a linear classifier and training the pipeline in the way of MAML. ALFA+fo-Proto-MAML (Baik et al., 2020) takes a further step to improve fo-Proto-MAML by adaptively generating the hyper-parameters (e.g. learning rate and weight decay coefficient) with a small meta-network. CNAPS (Requeima et al., 2019) introduces FiLM (Perez et al., 2018) module to adapt the parameters of embedding network and classifier to the new tasks. Similar work also includes (Tseng et al., 2020). Further, Simple CNAPS (Bateni et al., 2020) is proposed to replace the parametric classifier with a class-covariance-based distance metric. In contrast, from the perspective of 'pre-training' pipeline, SUR (Dvornik et al., 2020) proposes to learn a coefficient vector to linearly combine the representations of several independent backbones that learned on specific datasets. Similarly, URT (Liu et al., 2021a) proposes a multi-head Universal Representation Transformer (Vaswani et al., 2017) layer to select features. Besides, FLUTE (Triantafillou et al., 2021) proposes to learn the convolutional layers as a universal template for all datasets while training BN layers specifically for each dataset. During test phase, a set of coefficient is outputted by a 'Blender Network' to linearly combine the pre-trained BN layers for prediction. Due to the several forward passes of all backbones, computational cost of SUR and URT during inference time is quite expensive. Thus, URL (Li et al., 2021) (see Fig. 7)proposes to distill a single powerful backbone from

the domain-specific backbones and simply train a classifier on top of the frozen backbone with few training samples of the new task during inference phase. As a variant of URL, TSA (Li et al., 2022) proposes to add a residual module to each $3 \times 3$ convolutional layer to learn better representations.

Our goal in this work resemble (Xu et al., 2020; Wang & Deng, 2021) which try to drop the unimportant style information like colors and preserve the robust and invariant features with good discrimination ability. However, compared with these methods, our proposed IFR is much simpler since we can complete the invariant-feature location and reconstruction with only a single attention head instead of performing a series of operations.

**Feature Reconstruction:** Feature reconstruction is widely applied in many previous works (Baker & Matthews, 2004; Cao et al., 2014; Sun et al., 2015), and recently attracts attention from few-shot classification tasks for learning specific representations. DeepEMD (Zhang et al., 2020) decomposes an image into a set of semantic representations and reconstructs the target image from the perspective of solving an optimal transport problem. FRN (Wertheimer et al., 2021) formulates the linear reconstruction of query features as an ridge regression problem with a closed-form solution. CrossAttention (Hou et al., 2019) projects the query features into the space of support data and makes predictions by measuring the distances of class-conditioned projections and targets. Similarly, CrossTransformer (Doersch et al., 2020) applies a Transformer based network to assemble *query-aligned* class prototypes for query data predictions.

## B DETAILS OF PRE-TRAINING PIPELINE IN CFC

Pre-training pipeline has attracted increasing attention from few-shot classification because of its impressive ability in extracting representations (Tian et al., 2020b; Dhillon et al., 2020). The pre-training pipeline firstly pre-trains a single embedding backbone with sufficient labeled samples. Then, during test phase, the backbone is frozen and a specific classifier is trained with limited support data pairs of a new task on top of the backbone to predict the labels of query data.

*Model Pre-training.* The goal of model pre-training is to obtain a powerful embedding encoder for feature extraction. In this paper, we follow URL (Li et al., 2021) to use a multi-domain model distilled from several domain-specific models as our backbone. The distillation process can be simply formulated as:

$$\min_{\phi,\psi_i} \sum_{i=1}^{|\mathcal{S}^{\mathrm{seen}}|} \frac{1}{|\mathcal{D}_i^{\mathrm{tr}}|} \sum_{\boldsymbol{x},y \in \mathcal{D}_i^{\mathrm{tr}}} \Big( \ell(g_{\psi_i} \circ f_\phi(\boldsymbol{x}), y) + \lambda \mathcal{R}(\phi, \psi_i) \Big),$$

where $\mathcal{D}_i^{\mathrm{tr}}$ is the training set of the $i$-th sub-dataset, $\ell(\cdot)$ is the cross-entropy loss function, $f_\phi$ is the backbone parameterized with $\phi$, $g_{\psi_i}$ is the classifier for domain $i$ and parameterized with $\psi_i$. $\mathcal{R}$ is the regularization loss and $\lambda$ is the coefficient of the regularization loss. In URL (Li et al., 2021), the regularization loss is composed of two losses that aim to minimize the distance between the predictions and maximize the similarity of features between the distilled model and the corresponding domain-specific model.

*Classifier learning.* Since there exists gaps among domains, it is difficult for a pre-trained model to extract features that can generalize to previously unseen domains. An efficient solution for this problem is learning a linear transformation to project features learned from the pre-trained backbone into the task-specific space (Li et al., 2021).

We denote the transformation module as $h_\theta$ which is parameterized with $\theta$. Given a support set $\mathcal{D}_{\mathcal{T}^{\mathrm{new}}} = \{(\boldsymbol{x}_j^{\mathrm{s}}, y_j^{\mathrm{s}})\}_{j=1}^{|\mathcal{D}_{\mathcal{T}^{\mathrm{new}}}|}$ of a new task, following Mensink et al. (2013); Snell et al. (2017) and Dvornik et al. (2020), the centroid of each class is calculated firstly by averaging the corresponding support features extracted from the pre-trained backbone $f_{\phi^*}$ and transformation module $h_\theta$:

$$\boldsymbol{c}_i = \frac{1}{|\mathcal{C}_i|} \sum_{\boldsymbol{x}^{\mathrm{s}} \in \mathcal{C}_i} h_\theta \circ f_{\phi^*}(\boldsymbol{x}^{\mathrm{s}}),$$

$$\mathcal{C}_i = \{\boldsymbol{x}_j^{\mathrm{s}} | y_j^{\mathrm{s}} = i\}, i = 1, 2, ..., N_c, \tag{6}$$

where $\phi^*$ denotes the optimal backbone parameters, $N_c$ is the number of classes in given support set $\mathcal{D}_{\mathcal{T}^{\mathrm{new}}}$. Then, the label of each given support data point $\boldsymbol{x}^{\mathrm{s}} \in \mathcal{D}_{\mathcal{T}^{\mathrm{new}}}$ is estimated via measuring the

distances between the data point and centroids:

$$p(\hat{y} = t | \boldsymbol{x}^{\mathrm{s}}) = \frac{e^{\cos(h_\theta \circ f_{\phi^*}(\boldsymbol{x}^{\mathrm{s}}), \boldsymbol{c}_t)}}{\sum_{l=1}^{N_c} e^{\cos(h_\theta \circ f_{\phi^*}(\boldsymbol{x}^{\mathrm{s}}), \boldsymbol{c}_l)}}, \tag{7}$$

where $\hat{y}$ denotes the prediction and $\cos(\cdot, \cdot)$ denotes the cosine similarity function. By solving the empirical risk minimization problem on the given support set $\mathcal{D}_{\mathcal{T}^{\mathrm{new}}}$:

$$\min_\theta \frac{1}{|\mathcal{D}_{\mathcal{T}^{\mathrm{new}}}|} \sum_{\boldsymbol{x}_j^{\mathrm{s}}, y_j^{\mathrm{s}} \in \mathcal{D}_{\mathcal{T}^{\mathrm{new}}}} \log\left(p(\hat{y} = y_j^{\mathrm{s}} | \boldsymbol{x}_j^{\mathrm{s}})\right), \tag{8}$$

the optimal parameters $\theta^*$ are obtained. Then, the likelihood of query data can be estimate with $\theta^*$ by performing 7.

## C    DATASET

Meta-Dataset (Triantafillou et al., 2020) is proposed as a few-shot classification benchmark originally with only 10 datasets: ILSVRC_2012 (a.k.a ImageNet) (Russakovsky et al., 2015), Omniglot (Lake et al., 2015), FGVC-Aircraft (a.k.a Aircraft) (Maji et al., 2013), CUB-200-2011 (a.k.a CU Birds) (Wah et al., 2011), Describable Textures (a.k.a DTD) (Cimpoi et al., 2014), QuickDraw (Jongejan et al., 2016), FGVCx Fungi (a.k.a Fungi) (Schroeder & Cui, 2018), VGG_Flower (Nilsback & Zisserman, 2008), Traffic Sign (Houben et al., 2013), MSCOCO (Lin et al., 2014). Later, MNIST (LeCun et al., 1998), CIFAR-10 (Krizhevsky et al., 2009) and CIFAR-100 (Krizhevsky et al., 2009) are included.

According to Triantafillou et al. (2020), there are two kinds of experimental settings: *'Training on all datasets'* and *'Training on ImageNet only'*. The difference of the two settings is how many datasets are accessible to the model during meta-training phase. In 'Training on all datasets' setting, the training sets of the first 8 datasets(ImageNet, Omniglot, Aircraft, CU Birds, DTD, QuickDraw, Fungi and VGG_Flower) are treated as training set where the training tasks are sampled. In 'Training on ImageNet only' setting, only the training set of ILSVRC_2012 is accessible to the model. During meta-test phase, the model is evaluated on 600 tasks each randomly sampled from the test sets of accessible datasets and the remaining datasets(Traffic Sign, MSCOCO, MNIST, CIFAR-10 and CIFAR-100). The tasks in both meta-training and meta-test phases are sampled with varying number of ways and shots as proposed in Triantafillou et al. (2020).

## D    MORE IMPLEMENTATION DETAILS

### D.1    TRAINING BACKBONES

In this paper, we follow URL (Li et al., 2021), which transforms the universal representations extracted from a frozen pre-trained backbone to the tasks-specific space for classification, to train a specific IFR module on top of the pre-trained backbone for each single task. The domain-specific backbones are trained in typical supervised learning paradigm on specific dataset respectively while the multi-domain backbone is distilled from 8 domain-specific backbones.

For both learning processes of domain-specific backbones and the multi-domain backbone, the models take a batch of data and labels for training as done in typical supervised learning while are evaluated for validation performance on few-shot classification tasks sampled from corresponding validation sets. Follow the protocal in Triantafillou et al. (2020), the training episodes have 50% probability coming from ImageNet dataset during the distillation of the multi-domain backbone.

For more concrete details about hyper-parameters, please refer to Li et al. (2021).

### D.2    TRAINING IFR MODULE

The *Invariant-content Feature Reconstruction* (IFR) module proposed in this paper is a shallow neural network in order to simultaneously consider both high-level and fine-grained invariant-content features. The IFR module is composed of a single attention head and a linear transformation head.

---

**Algorithm 1** Invariant-content Feature Reconstruction Algorithm

---

**Input:** pre-trained backbone $f_{\phi^*}$, number of inner iterations $n$, learning rate $\eta$, query head parameters $A_{\theta_{\mathrm{q}}}$, key head parameters $A_{\theta_{\mathrm{k}}}$, value head parameters $A_{\theta_{\mathrm{v}}}$, linear transformation parameters $h_\theta$. Let $\boldsymbol{\Theta} = \{\theta_{\mathrm{q}}, \theta_{\mathrm{k}}, \theta_{\mathrm{v}}, \theta\}$.

**Output:** the optimal parameters of query head $\theta_{\mathrm{q}}^*$, key head $\theta_{\mathrm{k}}^*$, value head $\theta_{\mathrm{v}}^*$ and linear transformation head $\theta^*$.

*# Augmented task generation*

**Sample** a new task $\mathcal{T} = \{\{X^{\mathrm{s}}, Y^{\mathrm{s}}\}, \{X^{\mathrm{q}}, Y^{\mathrm{q}}\}\}$;

**Generate** the augmented support data $\{X^{\mathrm{a}}, Y^{\mathrm{a}}\}$;

*# Training on support data*

**for** $i = 1$ **to** $n$ **do**

    **Compute** queries, keys and values:

        $\boldsymbol{Q} = A_{\theta_{\mathrm{q}}} \circ f_{\phi^*}(X^{\mathrm{s}})$, $\boldsymbol{K} = A_{\theta_{\mathrm{k}}} \circ f_{\phi^*}(X^{\mathrm{aug}})$, $\boldsymbol{V} = A_{\theta_{\mathrm{v}}} \circ f_{\phi^*}(X^{\mathrm{aug}})$;

    **Flatten** $\boldsymbol{Q}, \boldsymbol{K}, \boldsymbol{V}$ to $\bar{\boldsymbol{Q}}, \bar{\boldsymbol{K}}, \bar{\boldsymbol{V}}$;

    *# Invariant-content feature reconstruction*

    **Reconstruct** invariant-content features $\hat{X}^{\mathrm{s}}$:

        $\hat{X}^{\mathrm{s}} = softmax(\frac{\bar{\boldsymbol{Q}} \cdot \bar{\boldsymbol{K}}^\top}{\sqrt{c}}) \cdot \bar{\boldsymbol{V}}$;

    *# Feature Fusion*

    **Combine** the high-level and invariant-content features:

        $Z^{\mathrm{s}} = h_\theta(f_{\phi^*}(X^{\mathrm{s}}) + \alpha \cdot \hat{X}^{\mathrm{s}})$;

    **Update** parameters:

        $\boldsymbol{\Theta} \leftarrow \boldsymbol{\Theta} - \eta \nabla_{\boldsymbol{\Theta}} \mathbb{E}_{\boldsymbol{z}^{\mathrm{s}} \sim Z^{\mathrm{s}}} \log(p(\hat{y} = y^{\mathrm{s}} | \boldsymbol{z}^{\mathrm{s}}))$

**end for**

---

The attention head, which aims to capture the fine-grained invariant-content features, includes three linear layers which are respectively queries head, keys head and values head. In our IFR pipeline, the queries head takes the representations of original images, including both support and query data, as input while the keys and values heads take the representations of the augmented data as input.

Inspired by the idea that pixels, which contains invariant content information, in original images ought to highly resemble the corresponding parts in the augmented data, we approximately extract the invariant-content features via retrieving content features that are invariant to style modifications from the augmented data by measuring the similarities of features in pixel level. To this end, in IFR, we simulate the style modifications with content-preserving data augmentations. By treating original support and query data as queries and the augmented support data as keys and values, we can firstly explore the invariant-content features via measuring the similarities among pixels, and then retrieving the features via combining the features with similarity weights. Since content features are invariant to style modifications, the similarity of two content feature pixel will be relatively larger.

The other part in IFR module is a linear transformation head which is composed of a batch normalization layer, an average pooling layer and a linear layer. After the invariant-content features are reconstructed, we further combine them with the high-level semantic features extracted from the frozen pre-trained backbone following Eq. (2) and map the feature fusion to the task-specific space.

In our method, the IFR module is treated as a task-specific transformation in each task adaptation episode. Thus, at the beginning of each episode, the IFR module is re-initialized with a specific way. To be specific, all linear layers of the IFR module are re-initialized with the identity matrices. In particular, a scale $10^{-4}$ is multiplied to the weights of linear layers of the attention head. Besides, the weights and biases of Batch Normalization layer are re-initialized with zeros and ones.

The adaptation for each task is realized by iteratively performing stochastic gradient descent on the loss with respect to fused support representations and labels. Each adaptation episode consists of 50 iteration steps. The optimizer used in our experiments is Adadelta (Zeiler, 2012). We set the learning rate to 1.0 for "Traffic Sign" and "MNIST" datasets while 0.1 for the remaining datasets.

We evaluate our method on 600 randomly sampled tasks for each candidate dataset, and report the average accuracy and the 95% confidence score as the results in all experiments. To manifest the stability of our method and avoid extremely good or bad results, we run all experiments in this paper 10 times with different random seeds. The detailed algorithm is reported in Algoritim 1.

### D.3 DATA AUGMENTATION

Data augmentation plays an important role in our proposed IFR method. We use data augmentations to simulate content-preserving style modifications and further explore the invariant-content features that are insensitive to style changes. The assumptions behind this derive from contrastive learning (Chen et al., 2020; He et al., 2020; Grill et al., 2020; Mitrovic et al., 2020). In the context of contrastive learning, it is assumed that images are generated by two essential elements, i.e. content and style, and the fundamental content information is independent of style information like background and colors (Mitrovic et al., 2020). Inspired by these assumptions, we expect that fine-grained invariant-content features can be captured by retrieving content features that are robust to style changes from a set of augmented data. Concretely, in this paper, we select `RandomSizedCrop`, `RandomGrayscale`, `RandomHorizontalFlip` and `RandomColorJitter` as content-preserving data augmentations.

**RandomSizedCrop.** Randomsizedcrop augmentation crops a portion of an image along the height and width of the image and then resizes the image to a given size. In our experimental settings, we set the portion interval as $[0.75, 1.0]$ and the size as $(84, 84)$ which means $0 \sim 25\%$ of the area of a given image will be cropped and the preserved image will then be reized to $84 \times 84$.

**RandomGrayscale.** Randomgrayscale augmentation randomly convert an image to grayscale with a given probability. In our experimental settings, we set the probability to $20\%$.

**RandomHorizontalFlip.** Randomhorizontalflip randomly flips an image along the horizontal direction with a probability. In our experimental settings, we set the probability to $50\%$.

**RandomColorJitter.** Randomcolorjitter augmentation randomly modifies the brightness, contrast, saturation and hue of an image with specific given probabilities. We set the scales of the modifications to $5\%$ for all aspects and the probability of conducting this augmentation to $30\%$. Thus, if a decision is made to change the style of an image, the brightness, contrast and saturation will be modified with a factor that is chosen uniformly from $[max(0, 0.95), 1.05]$ while the hue will be modified with a factor that is uniformly chosen from $[-0.05, 0.05]$.

## E DETAILED EXPERIMENTAL RESULTS.

### E.1 COMPLETE RESULTS OF TRAINING ON ALL DATASETS

We respectively report our results and learning curves under 'Train on all datasets' settings in Table 1 and Fig. 11. We compare our method with the existing state-of-the-art methods, such as Proto-MAML (Triantafillou et al., 2020), CNAPS (Requeima et al., 2019), SimpleCNAPS (Bateni et al., 2020), SUR (Dvornik et al., 2020), URT (Liu et al., 2021a), FLUTE (Triantafillou et al., 2021), Tri-M (Liu et al., 2021b), 2LM (Qin et al., 2023) and URL (Li et al., 2021). Among all methods, IFR achieves the best performance in average and ranks 2.4. The average performance of our method is comparably with URL (Li et al., 2021) on seen datasets while better than all other methods on unseen datasets.

Compared with URL, which performs better than other methods except ours in average, IFR obtains better results on Omniglot (+0.2%), Textures (+0.2%), QuickDraw (+0.3%), VGG Flower (+0.8%), Traffic Sign (+3.7%), MSCOCO (+1.7%), MNIST (+0.6%), CIFAR-10 (+1.1%) and CIFAR-100 (+1.0%). The results show that our method can achieve better generalization performance on unseen domains, which is more challenging than seen domains. Since there exists discrepancies among the distributions of datasets, the backbone can only recognize and extract semantic information that shared across seen and unseen domains while fail to explore the representative features. IFR addresses this problem by additionally extracting the specific invariant-content features from original data and their augmented counterpart. Besides, we also notice that IFR outperforms recent work, 2LM (Qin et al., 2023), on 7 out of 13 datasets, and achieves better average performance on both seen and unseen domains.

Although IFR achieves good performance, we still observe that the performance of IFR drops on ImageNet (-0.3%) and Fungi (-1.1%) datasets compared with URL. Intuitively, it is expected that the performance will increase with the powerful representations extracted by the attention module. However, as shown in Fig. 11(a) and Fig. 11(g), the test accuracy curves of both ImageNet and

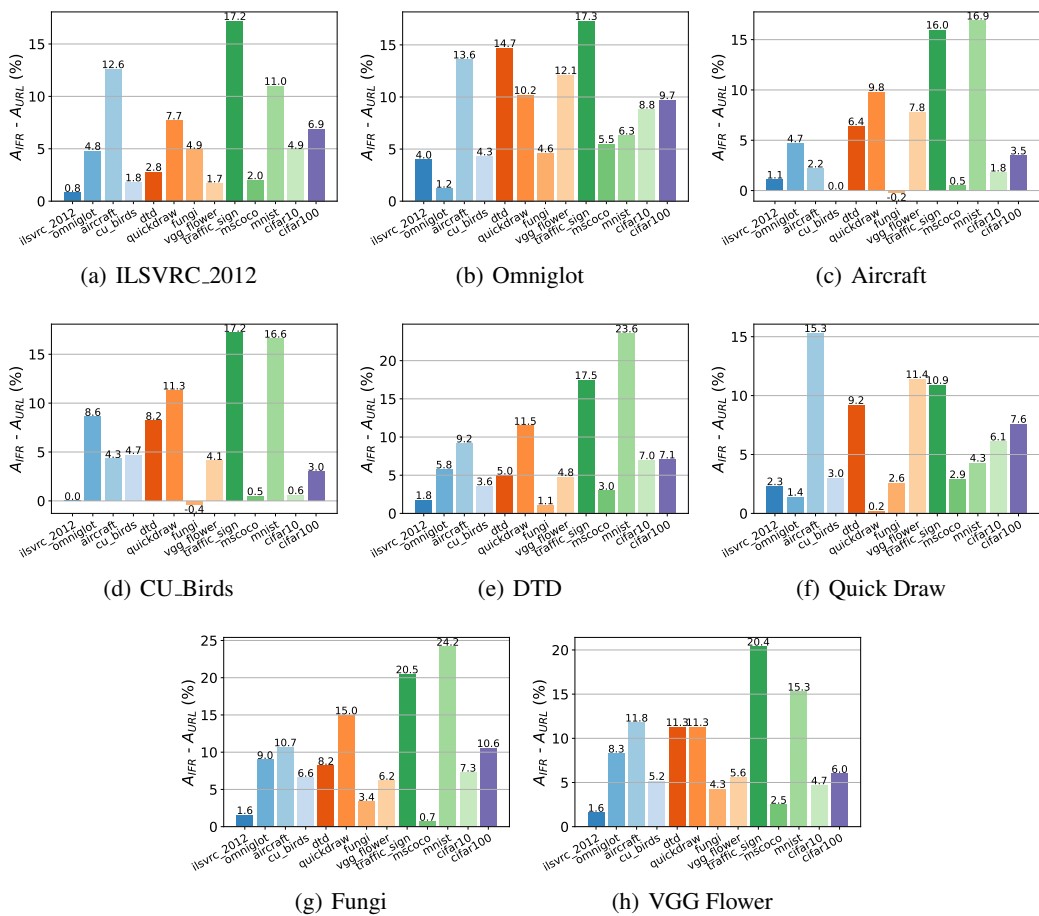

Figure 8: **Differences of test accuracies between IFR and URL on all datasets with different domain-specific backbones.** We evaluate our proposed IFR method on all domain-specifc backbones and report the differences of test accuracy between IFR and URL. As shown in figures, IFR outperforms URL with obvious gaps consistently on almost all backbones.

Fungi decline obviously after several adaptation steps, which is the typical phenomenon of overfitting. Such phenomenon also takes place slightly on MSCOCO dataset as shown in Fig. 11(j). For ImageNet, as proposed by Li et al. (2021); Triantafillou et al. (2020), the training episodes have 50% probability coming from ImageNet data source during pre-training phase. Thus, we conjecture that the pre-trained backbone has already owned the ability of extracting good representations from the ImageNet test data that share the same distribution with training data and the fused representations that contains fine-grained invariant-content features exacerbate the bias of the representations. On the other hand, compared with datasets like Aircraft in which the objects are evident, data samples in MSCOCO are more complicated since there are several objects in a single image. For example, the average number of objects is 7.7 for each image in MSCOCO. Thus, it is possible that IFR incorrectly captures the wrong semantic features and misclassifies the data with high confidence.

## E.2 TRAINING ON SINGLE BACKBONES

Following previous works (Triantafillou et al., 2020; Requeima et al., 2019; Bateni et al., 2020; Dvornik et al., 2020; Liu et al., 2021a; Triantafillou et al., 2021; Li et al., 2021), we evaluate our method under single domain few-shot classification settings respectively with 8 domain-specific backbones . A most popular case is evaluating the proposed method by performing classification on all datasets with ImageNet domain-specific backbone because ImageNet dataset includes huge number of images that can partially cover the distributions of other datasets like CU_Birds and VGG Flower etc.. The results evaluated on the backbone pre-trained on ImageNet are reported in Table 2.

Table 3: Results with different number of batches of augmented data. Mean accuracy, 95% confidence interval are reported. All results are the average of ten random reproduction experiments.

| Datasets | num=1 | num=2 | num=3 |
|---|---|---|---|
| ImageNet | 56.9±1.1 | **57.1±1.1** | 56.9±1.1 |
| Omniglot | 94.6±0.4 | 94.5±0.4 | 94.5±0.4 |
| Aircraft | **88.1±0.5** | 87.9±0.5 | **88.1±0.5** |
| Birds | 80.1±0.8 | 79.9±0.8 | **80.2 ±0.8** |
| Textures | **76.5±0.7** | 76.4±0.7 | 76.2±0.7 |
| QuickDraw | **82.4±0.6** | **82.4±0.6** | 82.2±0.6 |
| VGG Flower | **92.7±0.5** | 92.5±0.5 | **92.7±0.5** |
| Traffic Sign | 66.8±1.1 | 66.9±1.1 | **67.2±1.1** |
| MSCOCO | **55.9±1.0** | 55.6±1.0 | 55.6± 1.0 |
| MNIST | 95.2±0.4 | 95.2±0.4 | 95.2±0.4 |
| CIFAR-10 | 72.6±0.8 | 72.5±0.8 | 72.5±0.8 |
| CIFAR-100 | 64.0±1.0 | 64.0±1.0 | 63.9±1.0 |
| Average Seen | **79.8** | 79.7 | 79.7 |
| Average Unseen | **70.9** | 70.8 | **70.9** |
| Average All | **76.4** | 76.3 | 76.3 |

The results of "training on ImageNet only" show that IFR outperforms on 10 out of 13 datasets and ranks 1.4 in average. Compared with URL, our method consistently outperforms on all datasets with large gaps. Moreover, we also evaluate our method on other domain-specific backbones. The complete numerical results are available in Table 12. For convenience, we also provide a set of figures to depict the generalization performance differences between IFR and URL in Fig. 8. As revealed in figures, it is easy to find that IFR consistently facilitates to improve the generalization performance on almost all cases with large gaps, which demonstrates that learning fine-grained invariant-context features is necessary for new tasks that are sampled from unseen data and domains. For backbones pre-trained on simple datasets, such as the backbone learned from Omniglot, the improvements are significant, which indicate that the reconstructed invariant-content features play an important role in achieving better generalization performance when the prior knowledge of a backbone cannot cover the distributions of other domains.

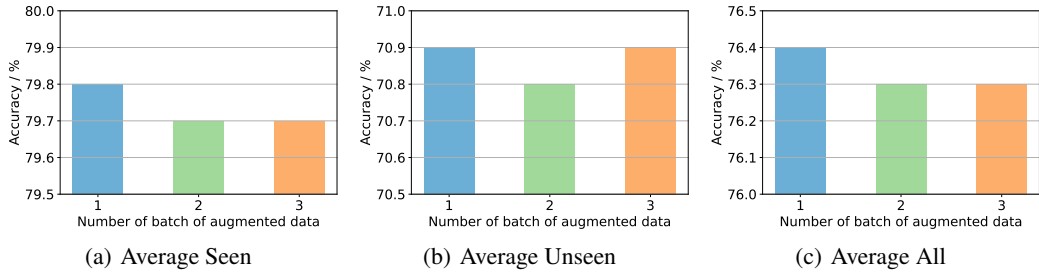

(a) Average Seen     (b) Average Unseen     (c) Average All

Figure 9: **Comparisons to the effect of different numbers of augmented data.** (a). Average performance of seen domains. (b). Average performance of unseen domains. (c). Average performance of all domains. As shown in three figures above, the average generalization performance on seen domains, unseen domains and all domains is not sensitive to the number of augmented data.

### E.3 EFFECT OF THE NUMBER OF BATCH OF AUGMENTED DATA

A reasonable intuition of IFR is that more augmented data facilitate to extract better invariant-content features and in turn achieve better generalization performance since more augmented data provide a set of more comprehensive semantic features. In order to figure out the effect of the number of augmented data on generalization performance, we evelute our proposed IFR method with different numbers of augmented data in this section. Specifically, we perform our IFR method under 'Train on all datasets' settings respectively with 1, 2 and 3 batches of augmented data.

Table 4: Comparisons to different scale coefficient. Mean accuracy, 95% confidence interval are reported. All results are the average of ten random reproduction experiments.

| Datasets | $\alpha = 1e-2$ | $\alpha = 1e-3$ | $\alpha = 1e-4$ | $\alpha = 1e-5$ |
|---|---|---|---|---|
| ImageNet | 56.6±1.1 | 56.9±1.1 | 56.9±1.1 | **57.2±1.1** |
| Omniglot | **94.7±0.4** | 94.5±0.4 | 94.6±0.4 | 94.6±0.4 |
| Aircraft | 88.2±0.5 | 88.0±0.5 | 88.1±0.5 | **88.3±0.5** |
| Birds | **80.3±0.8** | 80.0±0.7 | 80.1 ±0.8 | 80.1±0.8 |
| Textures | 76.3±0.7 | 76.4±0.7 | **76.5±0.7** | 76.4±0.7 |
| QuickDraw | 82.2±0.6 | **82.4±0.6** | **82.4±0.6** | **82.4±0.6** |
| Fungi | 66.8±1.0 | 66.6±1.2 | **66.9±1.0** | 66.8±1.0 |
| VGG Flower | 92.6±0.5 | **92.8±0.5** | 92.7±0.5 | 92.6±0.5 |
| Traffic Sign | 65.7±1.2 | **67.1±1.1** | 66.8±1.1 | 66.8±1.1 |
| MSCOCO | 55.8±1.0 | 55.6±1.0 | **55.9± 1.0** | 55.5±1.0 |
| MNIST | 95.0±0.5 | 95.1±0.4 | **95.2±0.4** | **95.2±0.4** |
| CIFAR-10 | 72.5±0.8 | **72.7±0.8** | 72.6±0.8 | 72.6±0.8 |
| CIFAR-100 | 63.6±1.0 | 63.9±1.0 | **64.0±1.0** | 63.7±1.0 |
| Average Seen | 79.7 | 79.7 | 79.8 | 79.8 |
| Average Unseen | 70.5 | **70.9** | **70.9** | 70.8 |
| Average All | 76.2 | 76.3 | **76.4** | 76.3 |

The experiment results are available in Fig. 9 and Table 3. It is easy to observe from Fig. 9 that the average generalization performance does not change obviously with the changes of the number of augmented data. However, according to the results reported in Table 3, different datasets react differently to the number of batches of augmented data. In detail, with the number of augmented data increasing, the performance on Traffic Sign slightly increases while the performance drops on Textures, QuickDraw, Fungi and MSCOCO.

## E.4 Effect of Scale Coefficient

As proposed in Eq. (2), before applying the linear transformation, the invariant-content features are combined with the high-level features extracted from the frozen pre-trained backbone with a scale coefficient. A simple case is directly combine the invariant-content and high-level features where the scale is 1.0. However, we find that the performance degrades significantly. Such phenomenon shows that directly applying the invariant-content features does not help improve the generalization performance and makes the learning process difficult. Thus, a key in IFR is finding a suitable scale coefficient for the fusion of high-level and fine-grained invariant-content features. To this end, in this section, we propose to explore the effect of scale coefficient on generalization performance.

Specifically, we perform our proposed IFR method under 'Train on all datasets' settings with different scale coefficients ranging from $1e-2$ to $1e-5$ to explore the effect of the scale coefficient. We report the experiment results in Fig. 6(b) and Table 4.

Fig. 6(b) plots the changes of the averaged test accuracy of all datasets with scale coefficient varying from $1e-2$ to $1e-5$. As shown in Fig. 6(b), our proposed IFR method outperforms URL baseline on all cases with an obvious gap. The averaged accuracy on all domains increases with the decrease of the scale coefficient and achieves the best performance when $\alpha = 0.0001$. When $\alpha$ becomes smaller, such as $\alpha = 0.00001$, the performance starts to drop. In detail, as shown in Table 4, different domains prefer different scale coefficients. For example, ImageNet, which provides more training episodes during pre-training phase, prefers smaller scale coefficient while CU_Birds, which owns evident objects and requires more fine-grained features, prefers the relatively larger scale coefficient. With all aspects taken into consideration, we choose $\alpha = 1e-4$ in our experiments.

## E.5 Effect of Initialization

In our experimental settings, we follow Li et al. (2021) and re-initialize the linear layers of our proposed IFR module with *identity matrix* at the beginning of each episode. There are two advantages in identity matrix initialization strategy. Firstly, the output of a linear layer that initial-

Table 5: Comparisons to IFR with different initialization. Mean accuracy, 95% confidence interval are reported. All results are the average of ten random reproduction experiments.

| Datasets | Identity | Random(Xavier) |
|---|---|---|
| ImageNet | 56.9±1.1 | **57.3±1.1** |
| Omniglot | 94.6±0.4 | 94.5±0.4 |
| Aircraft | 88.1±0.5 | 88.0±0.5 |
| Birds | 80.1±0.8 | **80.3±0.8** |
| Textures | **76.5±0.7** | 76.3±0.7 |
| QuickDraw | 82.4±0.6 | 82.3±0.6 |
| Fungi | **66.9±1.0** | 66.6±1.0 |
| VGG Flower | 92.7±0.5 | 92.6±0.5 |
| Traffic Sign | **66.8±1.1** | 66.5±1.1 |
| MSCOCO | **55.9±1.0** | 55.3±1.0 |
| MNIST | 95.2±0.4 | 95.2±0.4 |
| CIFAR-10 | **72.6±0.8** | 72.2±0.8 |
| CIFAR-100 | 64.0±1.0 | 63.9±1.0 |
| Average Seen | **79.8** | 79.7 |
| Average Unseen | **70.9** | 70.6 |
| Average All | **76.4** | 76.2 |

ized with identity matrix is the input itself. This helps make full use of the original representations extracted from the frozen pre-trained backbone. Moreover, initializing the parameters of IFR module as an identity matrix facilitates to optimize the IFR module from a good start point.

To further demonstrate the advantages of identity matrix initialization strategy, we evaluate our method respectively with *Xavier_Normal* (Glorot & Bengio, 2010) and identity matrix initialization strategies under 'Train on all datasets' settings. The results are reported in Table 5.

According to results reported in the Table 5, we observe that identity matrix initialization strategy outperforms Xavier_Normal slightly on most datasets. Specifically, IFR initialized with identity matrix achieves improvements on Textures (+0.2%), Fungi (+0.3%), Traffic Sign (+0.3%), MSCOCO (+0.6%) and CIFAR-10 (+0.4%). Compared with the results on seen domains, identity initialization performs better on unseen domains, which obtains 0.3% improvements in average.

To further demonstrate the advantage of applying identity matrix initialization strategy, we propose to compare the averaged test accuracies of 600 randomly sampled tasks after performing the first forward pass without any gradient descent update conducted on the parameters. As

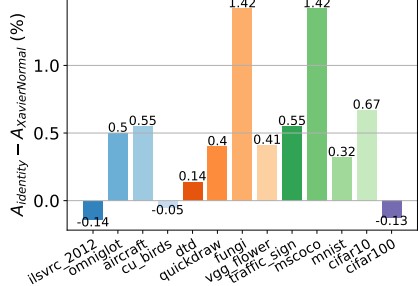

Figure 10: **Comparisons of accuracies after the first forward pass with different initialization.** The results show that identity matrix initialization can make full use of the original representations to achieve better performance.

shown in Fig. 10, the differences of accuracies manifest that IFR achieves better performance when directly using the original feature fusion for predictions, which is consistent with the aforementioned assumption that applying identity matrix initialization strategy helps make full use of the original representations learned from the pipeline and optimize model parameters from a good start.

### E.6   ABLATION STUDY ON BATCH NORMALIZATION

In our proposed method, a Batch Normalization layer is included in the linear transformation block. The main goal of the BN layer is to normalize the feature fusion that is obtained by combining high-level and fine-grained invariant-content features. Another effect that the BN layer imposes to the feature fusion is that the statistical information is collected and utilized for the predictions of the labels of query data. Since it has been demonstrated in previous works (Ioffe & Szegedy, 2015; Bronskill et al., 2020) that normalization techniques contribute to the stable training and conver-

Table 6: Comparison to our method with/without BN. Mean accuracy, 95% confidence interval are reported. All results are the average of ten random reproduction experiments.

| Datasets | URL | IFR w/o BN | IFR w BN |
|---|---|---|---|
| ImageNet | 57.2±1.1 | **57.4±1.1** | 56.9±1.1 |
| Omniglot | 94.3±0.4 | 94.3±0.4 | **94.6±0.4** |
| Aircraft | 88.1±0.5 | **88.3±0.5** | 88.1±0.5 |
| Birds | 80.2±0.8 | **80.4±0.7** | 80.1 ±0.8 |
| Textures | 76.3±0.7 | **76.5±0.7** | **76.5±0.7** |
| QuickDraw | 82.1±0.6 | 82.3±0.6 | **82.4±0.6** |
| Fungi | 68.0±1.0 | **68.1±1.0** | 66.9±1.0 |
| Traffic Sign | 63.1±1.1 | 63.7±1.1 | **66.8±1.1** |
| MSCOCO | 54.2±1.0 | 55.2±1.0 | **55.9± 1.0** |
| MNIST | 94.6±0.4 | 94.6±0.5 | **95.2±0.4** |
| CIFAR-10 | 71.5±0.8 | 72.0±0.8 | **72.5±0.8** |
| CIFAR-100 | 63.0±1.0 | 62.5±1.0 | **64.0±1.0** |
| Average Seen | 79.8 | 79.9 | 79.8 |
| Average Unseen | 69.3 | **69.6** | **70.9** |
| Average All | 75.7 | **75.9** | **76.4** |

Table 7: Ablation study on feature fusion approaches. Mean accuracy, 95% confidence interval are reported.

| Datasets | Sum | Average | Max | IFR |
|---|---|---|---|---|
| ImageNet | 52.6±1.2 | 52.6±1.2 | 54.4±1.1 | **56.9±1.1** |
| Omniglot | 94.4±0.4 | 94.3±0.4 | 94.1±0.4 | **94.6±0.4** |
| Aircraft | 87.0±0.5 | 87.0±0.5 | 87.7±0.5 | **88.1±0.5** |
| Birds | 76.5±0.8 | 77.0±0.8 | 79.1±0.8 | **80.1±0.8** |
| Textures | 74.4±0.7 | 74.4±0.7 | 76.2±0.7 | **76.5±0.7** |
| QuickDraw | 79.3±0.7 | 79.6±0.7 | 81.0±0.6 | **82.4±0.6** |
| Fungi | 58.6±1.1 | 58.3±1.2 | 64.0±1.1 | **66.9±1.0** |
| VGG Flower | 92.0±0.5 | 91.8±0.5 | 92.3±0.5 | **92.7±0.5** |
| Traffic Sign | 47.7±1.2 | 48.0±1.2 | 48.3±1.3 | **66.8±1.1** |
| MSCOCO | 48.4±1.0 | 48.8±1.1 | 48.1±1.0 | **55.9±1.0** |
| MNIST | 89.7±0.7 | 89.6±0.7 | 92.5±0.5 | **95.2±0.4** |
| CIFAR-10 | 70.0±0.8 | 69.7±0.8 | 70.1±0.8 | **72.6±0.8** |
| CIFAR-100 | 59.2±1.1 | 59.5±1.2 | 60.9±1.1 | **64.0±1.0** |
| Average Seen | 76.8 | 76.9 | 78.6 | **79.8** |
| Average Unseen | 63.0 | 63.1 | 64.0 | **70.9** |
| Average All | 71.5 | 71.6 | 73.0 | **76.4** |

gence which in turn help achieve better performance, we propose to explore the effect of the Batch Normalization layer imposes on features in the context of IFR. To be specific, we respectively evaluate our IFR method with and without the Batch Normalization layer under 'Train on all datasets' settings. The results are reported in Table 6 with mean accuracy and 95% confidence interval.

As we can observe from the table, IFR without BN performs better on seen domains. Compared with the case that IFR includes the BN, IFR without BN achieves 0.5%, 0.2%, 0.3% and 1.2% improvements respectively on ImageNet, Aircraft, CU_Birds and Fungi datasets. In contrast, IFR with BN reveals its impressive power on unseen datasets. Compared with the case that IFR excludes the BN, IFR with BN achieves 5.1%, 0.7%, 0.6%, 0.5% and 1.5% on unseen domains respectively. Although IFR without BN fails to outperform IFR with BN, it still outperforms URL baseline with 0.6%, 1.0% and 0.5% improvements respectively on Traffic Sign, MSCOCO and CIFAR-10 datasets. With all results taken into consideration, we find that plugging a BN in the linear transformation block indeed facilitates to improve the generalization performance, especially for unseen domains.

### E.7 ABLATION STUDY ON FEATURE FUSION APPROACHES

In our proposed IFR method, it is expected to take advantage of both invariant-content features and original high-level semantic features to obtain a set of representations that are informative and discriminative enough to achieve better generalization performance on Meta-Dataset. In this paper, the way that we choose to leverage these two sets of representations is simply adding the reconstructed fine-grained invariant-content features with a scale coefficient to the high-level original features. The reason for such operation is that the reconstructed invariant-content features are too strong and biased. If we directly add the reconstructed invariant-content features to the original features, overfitting will take place and the generalization performance will drop drastically.

Although the case we proposed in this paper has achieved good performance, we still want to conduct an ablation study to figure out whether other simple feature fusion approaches can achieve the same performance. Thus, in this section, we replace the feature fusion approach proposed in our method with 'Max', 'Sum' and 'Average' approaches to further figure out the problem mentioned above.

As shown in Table 7, it is easy to observe that the performance of candidate feature fusion approaches significantly degrade on all datasets compared with IFR. Among all candidate feature fusion approaches, we can observe that 'Max' fusion approach achieves the best results.

### E.8 ABLATION STUDY ON AUGMENTATIONS

In our proposed IFR method, a set of augmentations, including cropping, flipping, grayscaling and color jittering, is adopted as content-preserving augmentations in order to stimulate the case that the style information of images are modified while the content are well preserved. In this section, in order to have a comprehensive understanding of the effect of these augmentations, we perform an ablation study on all augmentations adopted in our method to figure out what roles they play in achieving good generalization performance. The results are reported in Table 8.

Generally, according to the results reported in Table 8, it is easy to observe that different datatsets prefer different augmentations. Take ImageNet as an example, the performance increases when cropping is removed and drop when color jitteing is removed. We conjecture the reason of such phenomenon is that cropping may cut some information of the main semantic features in images and in turn make it much more difficult to learn precise representations of target classes. Moreover, since color jittering helps modify the color of images, it facilitates to make the model to learn the robust and discriminative features by comparing the original images and their modified couterparts.

In addition, although we can observe that removing some augmentation techniques contributes to improving the performance of IFR on some datasets, we still choose to adopt all of them in our framework due to the better averaged performance on seen and unseen domains.

Further, in order to explore the effect of augmentations, we conduct an ablation study under 'Train on all datasets' settings. We replace the augmented data, which are used as keys and values, in our method with original data. The results are reported in Table 9. According to the table, IFR with augmentations achieves relatively better results compared with that without augmentations. Specifically, augmentations are important for datasets like ImageNet, Fungi, Traffic Sign, MSCOCO and CIFAR100. We conjecture the reason for such phenomenon is that augmentations helps these datasets which owns more objects and style information better capture the key representations.

### E.9 ABLATION STUDY ON ATTENTION MODULE

In our proposed IFR method, a single attention head is adopted to perform feature reconstruction in a fine-grained way by first measuring the similarities among representations between original and augmented data to locate the invariant-content features, and then highlighting these invariant-content features with large weights. In order to figure out the effect of the attention module in IFR, we conduct an ablation study regarding the attention head in this section. To be specific, we remove the attention module and directly combine the features of original and augmented data with simpler ways, including 'SUM', 'AVG' and 'MAX'. The results are reported in Table 10.

According to the results reported in the table, on the one hand, we find that 'AVG' and 'MAX' achieve better generalization performance among all cases except our IFR with attention module.

Table 8: Ablation study on augmentations. Mean accuracy, 95% confidence interval are reported.

| Datasets | w/o Crop | w/o Colorjitter | w/o Grayscale | w/o Flip | IFR |
|---|---|---|---|---|---|
| ImageNet | **57.1±1.1** | 56.5±1.1 | 56.9±1.1 | 57.0±1.1 | 56.9±1.1 |
| Omniglot | 94.4±0.4 | 94.3±0.4 | 94.5±0.4 | 94.3±0.4 | **94.6±0.4** |
| Aircraft | **88.2±0.5** | 88.1±0.5 | 88.1±0.5 | **88.2±0.5** | 88.1±0.5 |
| Birds | 80.1±0.8 | 80.1±0.7 | 79.9±0.8 | **80.2±0.8** | 80.1±0.8 |
| Textures | 76.3±0.7 | 76.0±0.7 | 76.2±0.7 | 76.3±0.7 | **76.5±0.7** |
| QuickDraw | 82.2±0.6 | 82.3±0.6 | **82.6±0.6** | 82.4±0.6 | 82.4±0.6 |
| Fungi | 66.6±1.0 | 66.4±1.0 | 66.6±1.0 | 66.6±1.0 | **66.9±1.0** |
| VGG Flower | 92.4±0.5 | 92.6±0.5 | **92.8±0.5** | 92.6±0.5 | 92.7±0.5 |
| Traffic Sign | **66.9±1.1** | 66.7±1.1 | 66.6±1.1 | 66.5±1.1 | 66.8±1.1 |
| MSCOCO | 55.9±1.0 | **56.0±1.0** | 55.8±1.0 | **56.0±1.0** | 55.9±1.0 |
| MNIST | **95.2±0.4** | 95.1±0.5 | 95.0±0.4 | 95.0±0.4 | **95.2±0.4** |
| CIFAR-10 | 72.6±0.8 | **72.8±0.8** | 72.6±0.8 | 72.3±0.8 | 72.6±0.8 |
| CIFAR-100 | 63.7±1.0 | 63.7±1.0 | 64.0±1.0 | **64.1±1.0** | 64.0±1.0 |
| Average Seen | 79.7 | 79.5 | 79.7 | 79.7 | **79.8** |
| Average Unseen | 70.86 | 70.9 | 70.8 | 70.8 | **70.9** |
| Average All | 76.3 | 76.2 | 76.3 | 76.3 | **76.4** |

Table 9: Ablation study on all augmentations.

| Datasets | IFR w/o augs | IFR |
|---|---|---|
| ImageNet | 56.6±1.1 | **56.9±1.1** |
| Omniglot | 94.5±0.4 | **94.6±0.4** |
| Aircraft | 88.0±0.5 | **88.1±0.5** |
| Birds | **80.4±0.7** | 80.1±0.8 |
| Textures | **76.6±0.7** | 76.5±0.7 |
| QuickDraw | 82.2±0.6 | **82.4±0.6** |
| Fungi | 66.6±1.0 | **66.9±1.0** |
| VGG Flower | 92.6±0.5 | **92.7±0.5** |
| Traffic Sign | 66.5±1.1 | **66.8±1.1** |
| MSCOCO | 55.8±1.0 | **55.9±1.0** |
| MNIST | **95.3±0.4** | 95.2±0.4 |
| CIFAR-10 | **72.7±0.8** | 72.6±0.8 |
| CIFAR-100 | 63.7±1.0 | **64.0±1.0** |
| Average Seen | 79.7 | **79.8** |
| Average Unseen | 70.8 | **70.9** |
| Average All | 76.3 | **76.4** |

On the other hand, our IFR obtains the best results on all datasets, which shows that attention facilitates to capture better features. We conjecture that the reasons for such phenomenon mainly include two aspects. Firstly, attention owns the ability of extracting much more fine-grained features. Thus, the learned fine-grained invariant-content features are more informative and comprehensive than those obtrained via 'SUM', 'MAX' or 'AVG'. Moreover, when performing attention operation, the similarities among representations of original and augmented data are measured. This step is equivalent to evaluating the importance of features in pixel level and further helps highlight the most important features, i.e. invariant-content features.

### E.10 Running Time of IFR

In this section, we conduct a further experiment to compare the running time between IFR and URL. Specifically, we conduct this experiment on the same NVIDIA A100 GPU for fairness. As shown in Table 11, we find that the time that IFR consumes is about 4.8 times in average than that of URL. Since the calculation process of the attention head is conducted in pixel level, the process will be

Table 10: Ablation study on attention module. Mean accuracy, 95% confidence interval are reported.

| Datasets | SUM | AVG | MAX | IFR |
|---|---|---|---|---|
| ImageNet | 48.3±1.1 | 56.1±1.1 | 55.8±1.1 | **56.9±1.1** |
| Omniglot | 91.5±0.6 | 94.2±0.5 | 94.3±0.4 | **94.6±0.4** |
| Aircraft | 80.3±0.8 | 87.8±0.5 | 87.7±0.5 | **88.1±0.5** |
| Birds | 68.3±0.9 | 79.3±0.8 | 79.6±0.8 | **80.1±0.8** |
| Textures | 62.4±0.9 | 76.1±0.7 | 75.7±0.7 | **76.5±0.7** |
| QuickDraw | 75.8±0.8 | 81.9±0.6 | 82.0±0.6 | **82.4±0.6** |
| Fungi | 52.8±1.1 | 66.3±1.1 | 66.3±1.0 | **66.9±1.0** |
| VGG Flower | 81.3±0.8 | 92.3±0.5 | 92.3±0.5 | **92.7±0.5** |
| Traffic Sign | 47.4±1.3 | 66.1±1.1 | 66.7±1.3 | **66.8±1.1** |
| MSCOCO | 47.3±1.0 | 54.6±1.0 | 55.2±1.0 | **55.9±1.0** |
| MNIST | 87.9±0.7 | 95.0±0.5 | 94.8±0.5 | **95.2±0.4** |
| CIFAR-10 | 62.2±0.9 | 72.2±0.8 | 72.2±0.8 | **72.6±0.8** |
| CIFAR-100 | 48.9±1.1 | 63.4±1.0 | 63.0±1.1 | **64.0±1.0** |

Table 11: Comparison of running time between IFR and URL.

| Datasets | URL (s/iter) | IFR (s/iter) |
|---|---|---|
| ImageNet | 0.47 | 2.60 ($\times$ 5.5) |
| Omniglot | 0.50 | 1.95 ($\times$ 3.9) |
| Aircraft | 0.36 | 2.25 ($\times$ 6.3) |
| Birds | 0.54 | 2.06 ($\times$ 3.8) |
| Textures | 0.29 | 1.74 ($\times$ 6.0) |
| QuickDraw | 0.81 | 3.05 ($\times$ 3.8) |
| Fungi | 0.81 | 2.51 ($\times$ 3.1) |
| VGG Flower | 0.39 | 1.91 ($\times$ 4.9) |
| Traffic Sign | 0.71 | 3.01 ($\times$ 4.2) |
| MSCOCO | 0.71 | 3.11 ($\times$ 4.4) |
| MNIST | 0.34 | 2.21 ($\times$ 6.5) |
| CIFAR-10 | 0.34 | 2.27 ($\times$ 6.7) |
| CIFAR-100 | 0.83 | 2.96 ($\times$ 3.6) |

quite computationally expensive and time-consuming. Thus, it is reasonable that IFR spends much more time than URL.

### E.11 MORE VISUALIZATION RESULTS OF IFR

As aforementioned, IFR is able to consider both high-level and fine-grained invariant-content features simultaneously when performing cross-domain few-shot classification tasks so that more comprehensive and discriminative features can be extracted. To show such ability of IFR, we provide some representative visualization results in this section to compare the representations learned respectively with URL and our proposed IFR. As revealed in Fig. 12, 13, 14, 15, 16, 17, we are able to observe that IFR can learn more accurate and comprehensive representations compared with URL.

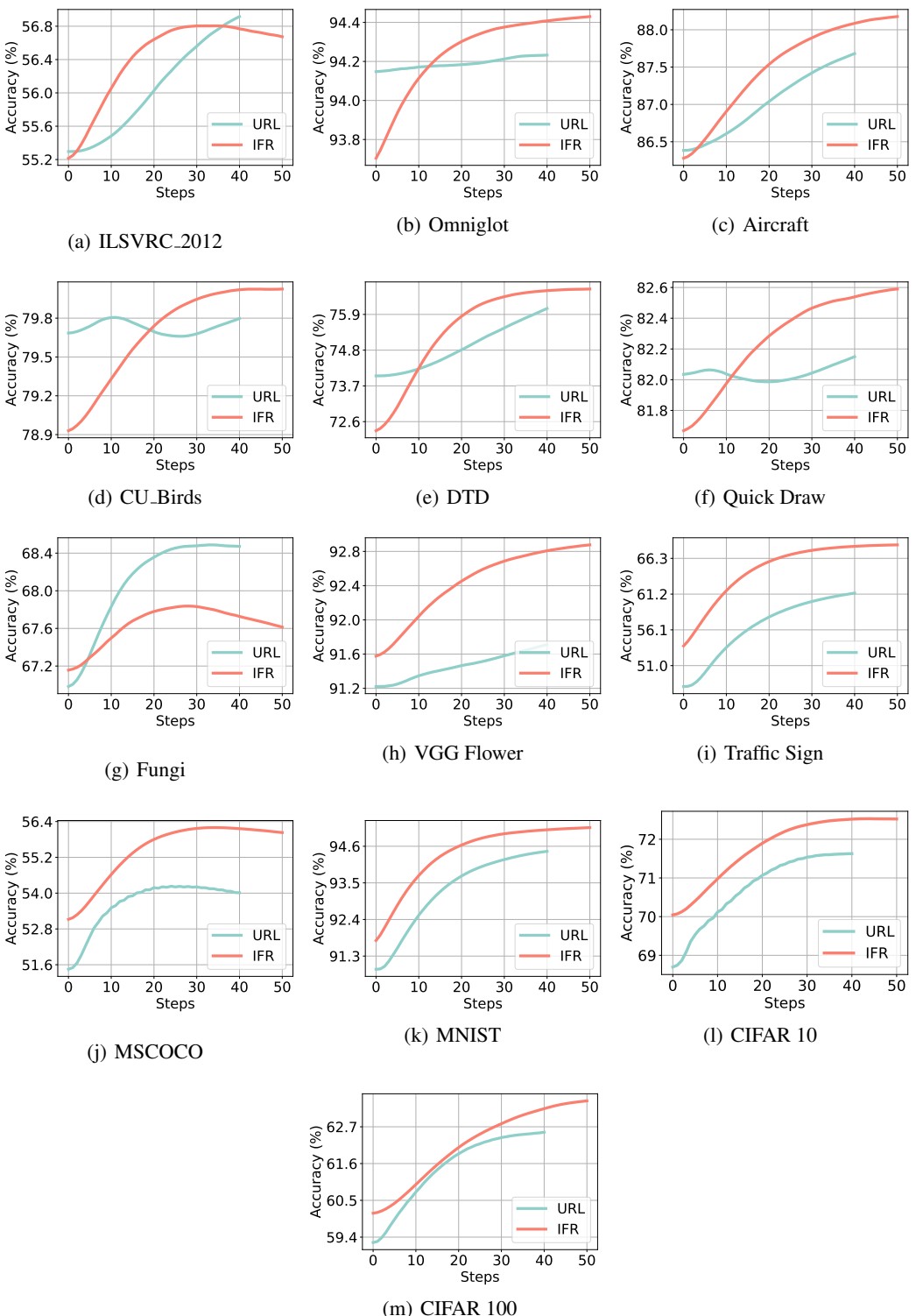

Figure 11: **Comparisons of test accuracy curves of IFR and URL on multi-domain backbone.**

Table 12: Numerical results of URL and IFR evaluated on single domain-specific backbones. Mean accuracy, 95% confidence interval are reported. All results of IFR are the average of ten random reproduction experiments.

| Datasets | Method | ILSVRC_2012 | Omniglot | Aircraft | CU_Birds | Textures | Quick Draw | Fungi | VGG Flower |
|---|---|---|---|---|---|---|---|---|---|
| ImageNet | URL | 55.8±1.0 | 17.1±0.6 | 21.7±0.7 | 25.4±0.8 | 24.2±0.8 | 24.1±0.8 | 32.9±0.9 | 25.0±0.8 |
| | IFR | 56.6±1.1 | 21.1±0.8 | 22.8±0.8 | 25.4±0.8 | 26.0±0.8 | 26.4±0.9 | 34.5±0.9 | 26.6±0.9 |
| Omniglot | URL | 67.4±1.2 | 93.2±0.5 | 58.2±1.2 | 58.7±1.4 | 57.3±1.4 | 78.4±1.0 | 57.6±1.3 | 54.6±1.3 |
| | IFR | 72.2±1.2 | 94.4±0.4 | 62.9±1.3 | 67.3±1.2 | 63.1±1.3 | 79.8±1.0 | 66.6±1.2 | 62.9±1.4 |
| Aircraft | URL | 49.5±0.9 | 16.8±0.5 | 85.7±0.5 | 31.4±0.8 | 26.0±0.7 | 23.8±0.6 | 31.0±0.7 | 24.6±0.6 |
| | IFR | 62.1±1.0 | 30.4±0.8 | 87.9±0.5 | 35.7±0.8 | 35.2±0.9 | 39.1±0.8 | 41.7±0.9 | 36.4±0.9 |
| Birds | URL | 71.2±0.9 | 13.0±0.6 | 19.9±0.7 | 65.0±0.9 | 19.6±0.7 | 16.7±0.7 | 42.8±1.0 | 28.9±0.8 |
| | IFR | 73.0±0.9 | 17.3±0.7 | 19.9±0.7 | 69.7±0.9 | 23.2±0.8 | 19.7±0.7 | 49.4±1.1 | 34.1±0.9 |
| Textures | URL | 73.0±0.6 | 25.0±0.5 | 38.6±0.7 | 42.2±0.7 | 54.9±0.7 | 38.6±0.6 | 54.1±0.7 | 43.2±0.7 |
| | IFR | 75.8±0.7 | 39.7±0.7 | 45.0±0.8 | 50.4±0.8 | 59.9±0.8 | 47.8±0.8 | 62.3±0.8 | 53.6±0.8 |
| QuickDraw | URL | 53.9±1.0 | 51.0±1.0 | 38.8±1.0 | 38.2±1.0 | 36.8±0.9 | 82.8±0.6 | 37.7±0.9 | 39.7±1.0 |
| | IFR | 61.6±1.0 | 61.2±1.0 | 48.6±1.1 | 49.5±1.1 | 48.3±1.1 | 83.0±0.6 | 52.7±1.1 | 51.0±1.1 |
| Fungi | URL | 41.6±1.0 | 9.1±0.5 | 14.9±0.7 | 25.5±0.8 | 15.6±0.7 | 12.5±0.6 | 65.8±0.9 | 23.3±0.8 |
| | IFR | 46.5±1.1 | 13.7±0.7 | 14.7±0.7 | 25.1±0.9 | 16.7±0.8 | 15.1±0.7 | 69.2±1.0 | 27.6±1.0 |
| VGG Flower | URL | 87.0±0.6 | 23.8±0.6 | 45.5±0.8 | 62.9±0.8 | 44.4±0.8 | 33.4±0.7 | 79.6±0.7 | 78.3±0.7 |
| | IFR | 88.7±0.6 | 35.9±0.8 | 53.3±0.9 | 67.0±0.9 | 49.2±0.9 | 44.8±0.9 | 85.8±0.7 | 83.9±0.6 |
| Traffic Sign | URL | 47.4±1.1 | 15.1±0.7 | 30.8±0.9 | 31.0±0.9 | 38.8±1.1 | 31.1±0.9 | 28.0±0.9 | 30.4±0.9 |
| | IFR | 64.6±1.1 | 32.4±1.0 | 46.8±1.1 | 48.2±1.1 | 56.3±1.2 | 42.0±1.1 | 48.5±1.2 | 50.8±1.1 |
| MSCOCO | URL | 53.5±1.0 | 12.9±0.6 | 22.5±0.8 | 25.1±0.9 | 23.7±0.8 | 21.3±0.8 | 32.5±1.0 | 25.7±0.8 |
| | IFR | 55.5±1.0 | 18.4±0.8 | 23.0±0.9 | 25.6±0.9 | 26.7±1.0 | 24.2±0.9 | 33.2±1.0 | 28.2±0.9 |
| MNIST | URL | 78.1±0.7 | 89.8±0.5 | 68.0±0.8 | 73.0±0.7 | 64.5±0.8 | 88.2±0.5 | 62.2±0.8 | 72.1±0.7 |
| | IFR | 89.1±0.7 | 96.1±0.4 | 84.9±0.8 | 89.6±0.6 | 88.1±0.8 | 92.5±0.5 | 86.4±0.7 | 87.4±0.8 |
| CIFAR-10 | URL | 67.3±0.8 | 28.5±0.6 | 41.2±0.7 | 41.8±0.8 | 36.9±0.7 | 40.0±0.7 | 38.8±0.7 | 41.3±0.8 |
| | IFR | 72.2±0.8 | 37.3±0.7 | 43.0±0.8 | 42.4±0.8 | 43.9±0.8 | 46.1±0.8 | 46.1±0.8 | 46.0±0.8 |
| CIFAR-100 | URL | 56.6±0.9 | 12.3±0.6 | 24.3±0.9 | 28.8±0.9 | 24.2±0.9 | 23.4±0.8 | 25.2±0.9 | 29.1±1.0 |
| | IFR | 63.5±1.0 | 22.0±0.9 | 27.8±1.0 | 31.8±1.0 | 31.3±1.0 | 31.0±1.0 | 35.8±1.1 | 35.1±1.1 |

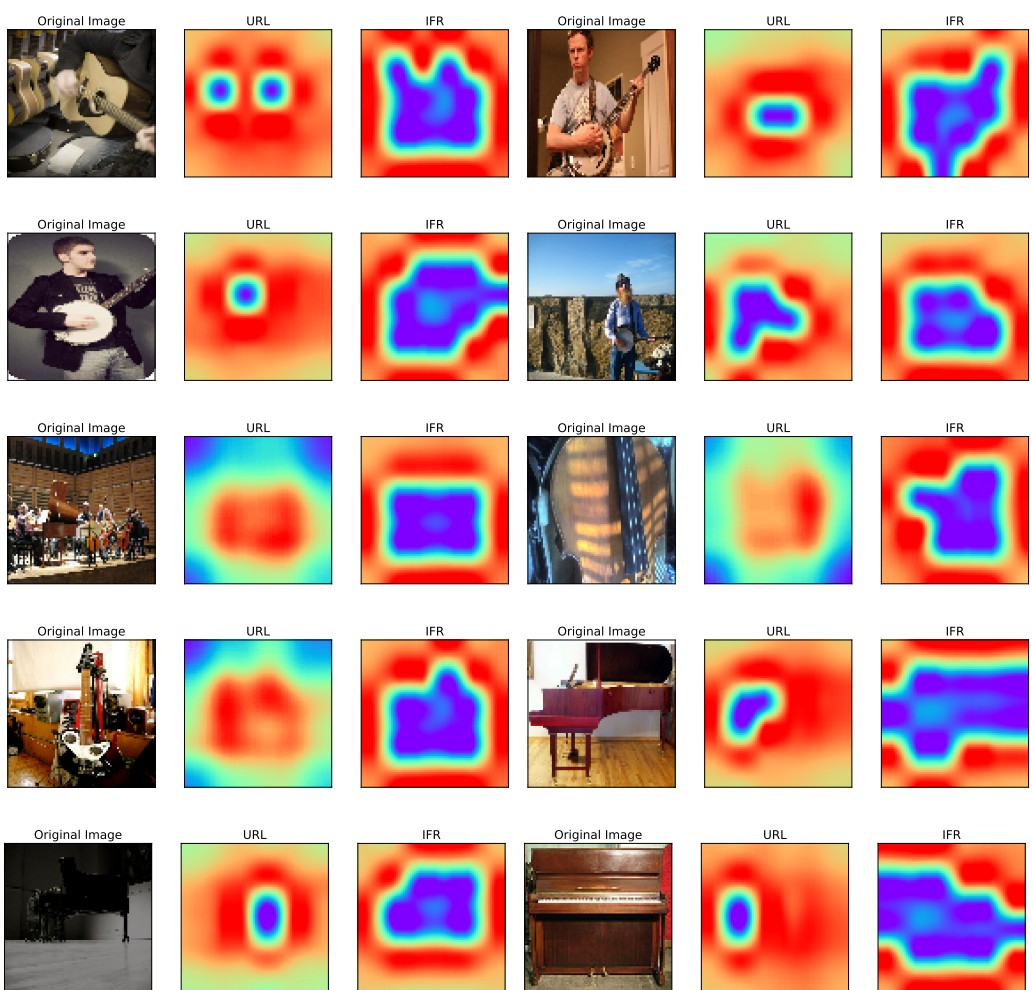

Figure 12: Comparions of features learned with URL and IFR on ImageNet.

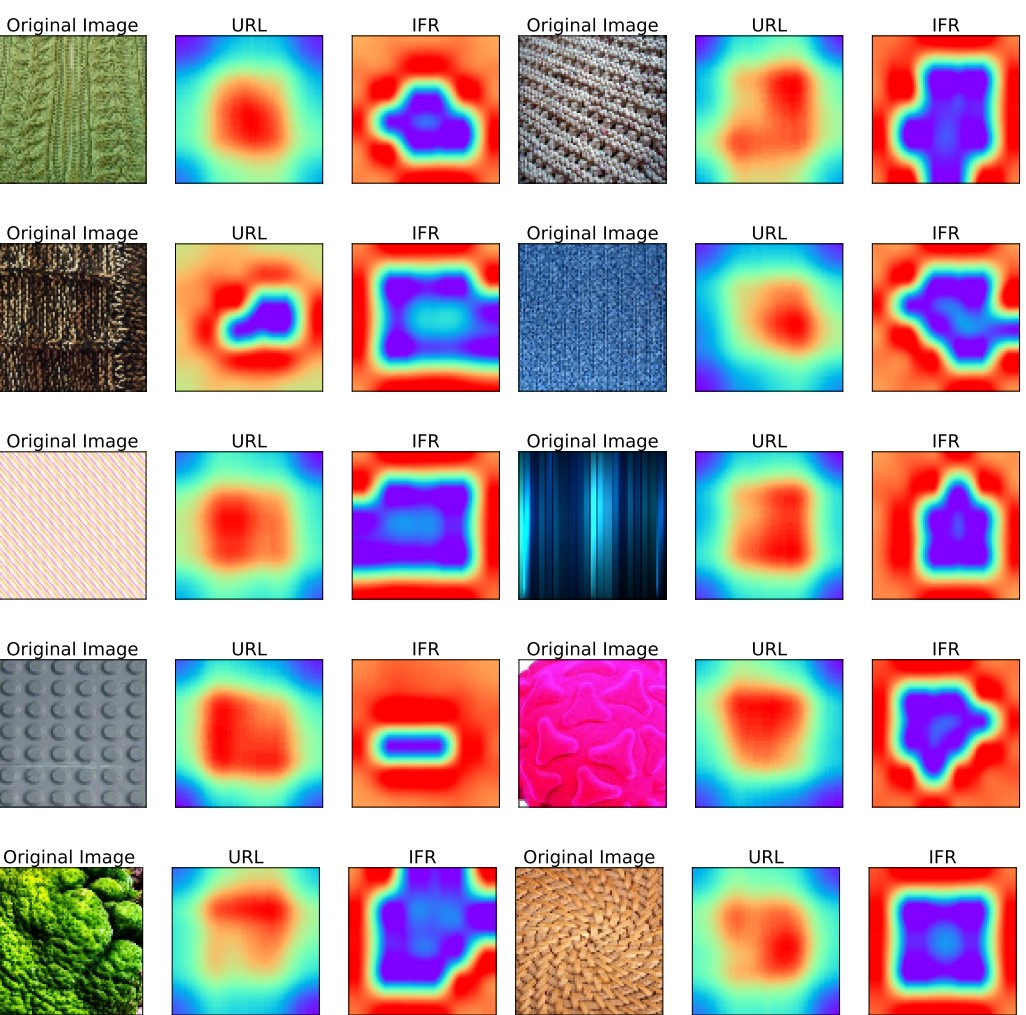

Figure 13: Comparions of features learned with URL and IFR on Textures.

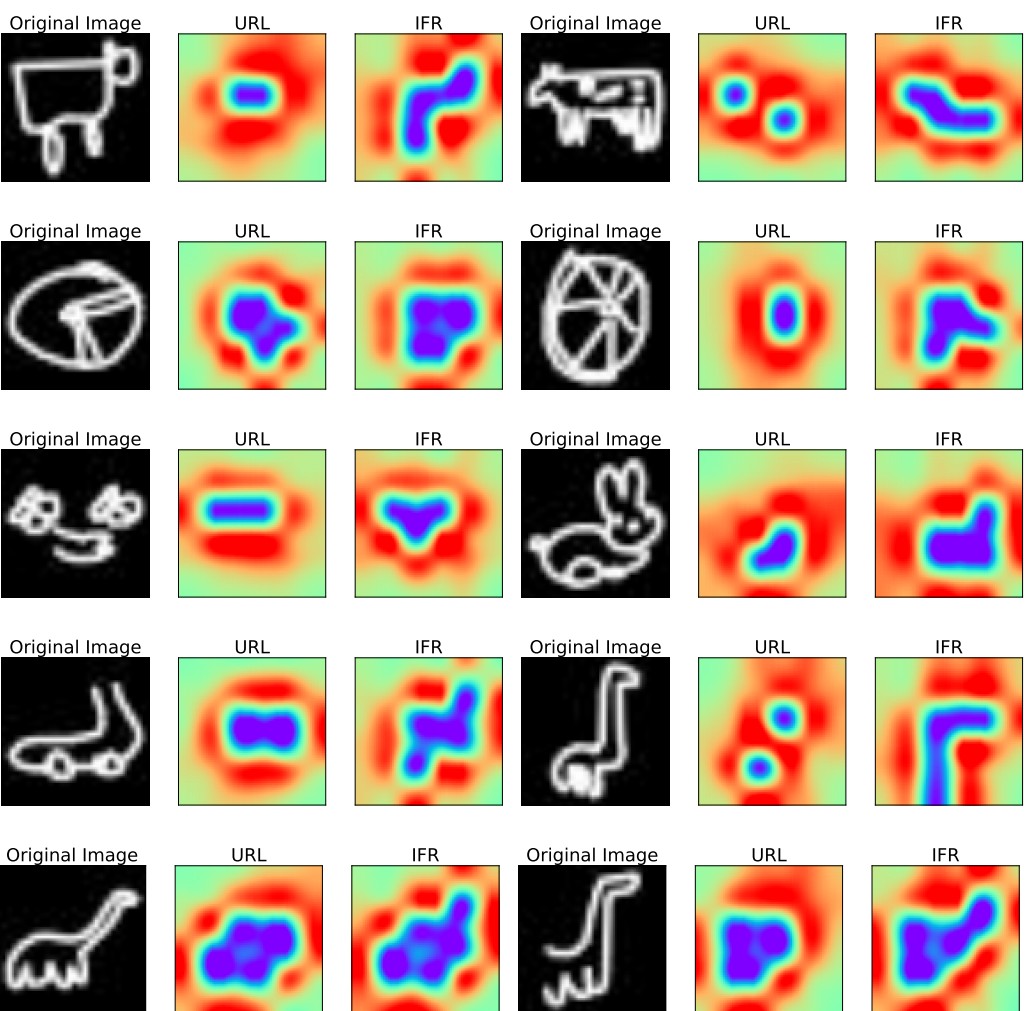

Figure 14: Comparions of features learned with URL and IFR on Quick Draw.

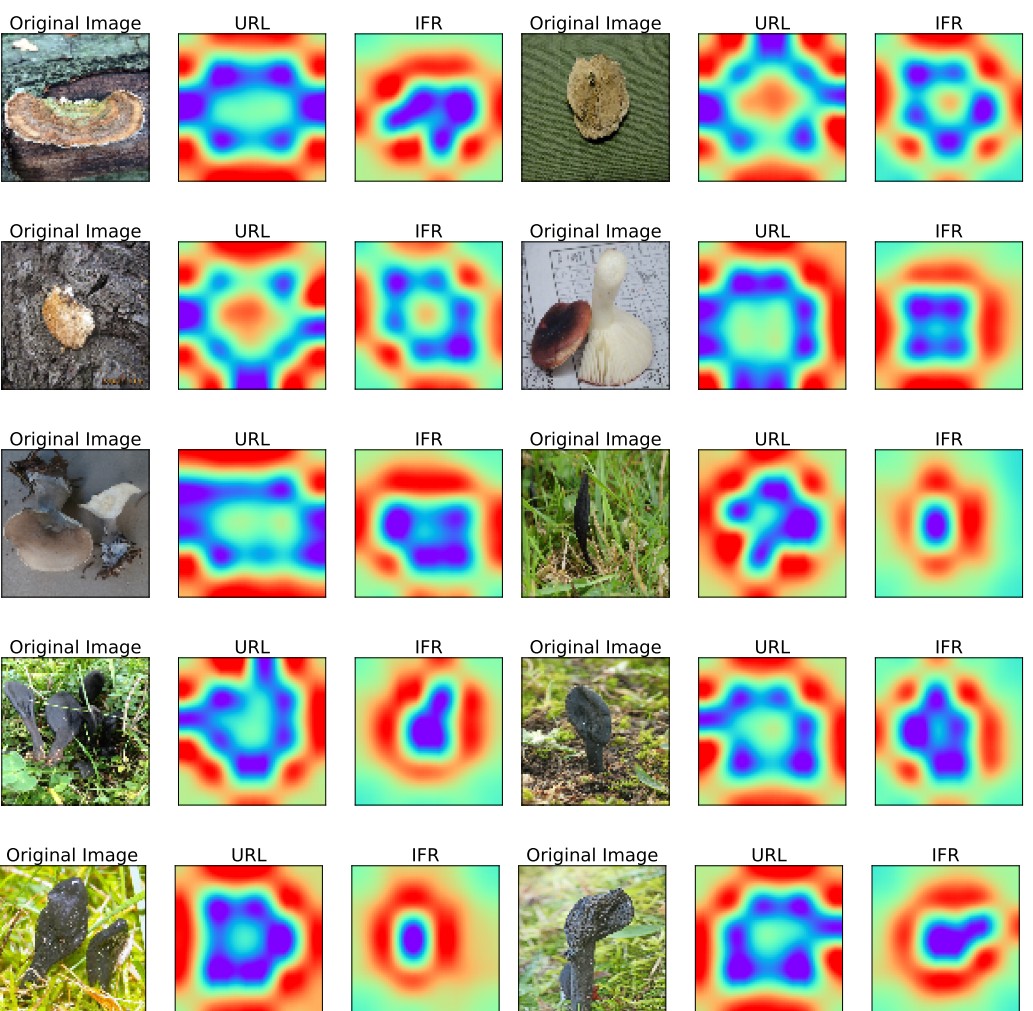

Figure 15: Comparions of features learned with URL and IFR on Fungi.

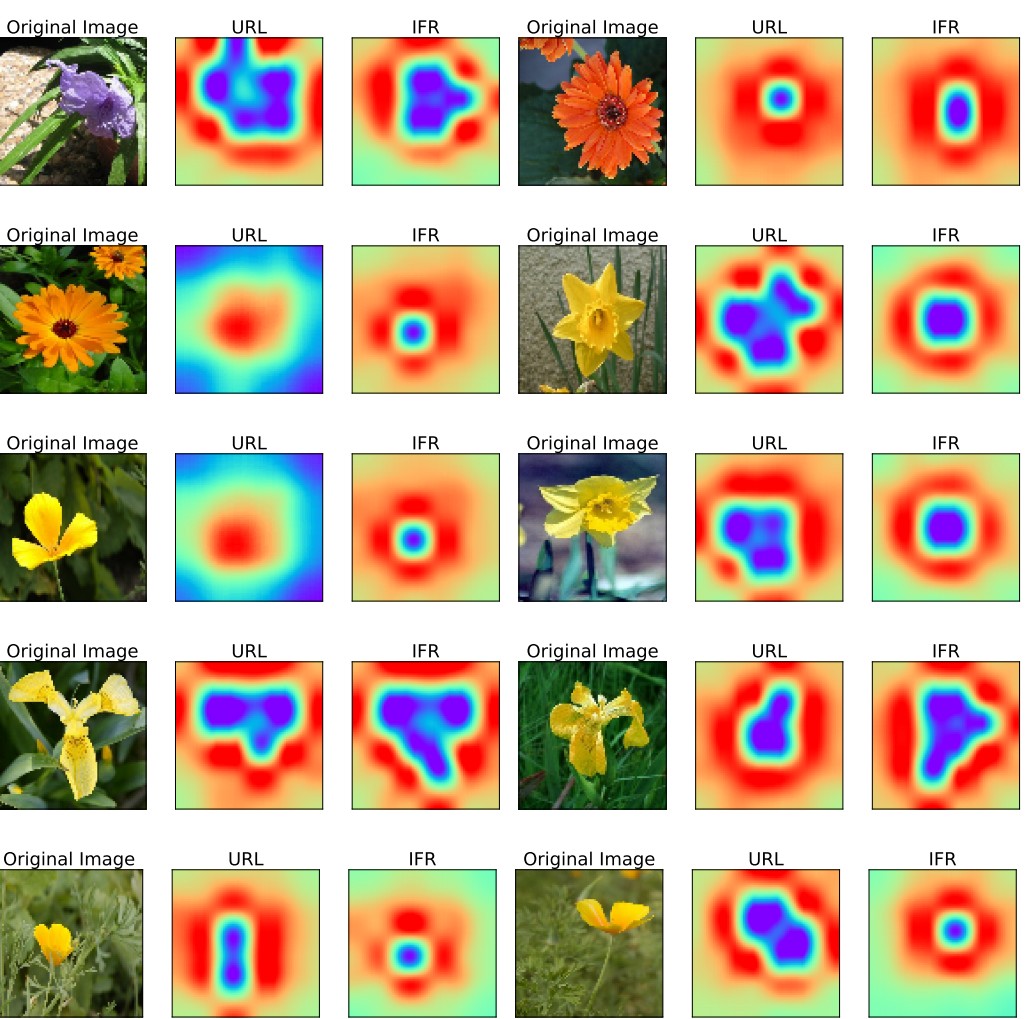

Figure 16: Comparions of features learned with URL and IFR on VGG_Flowers.

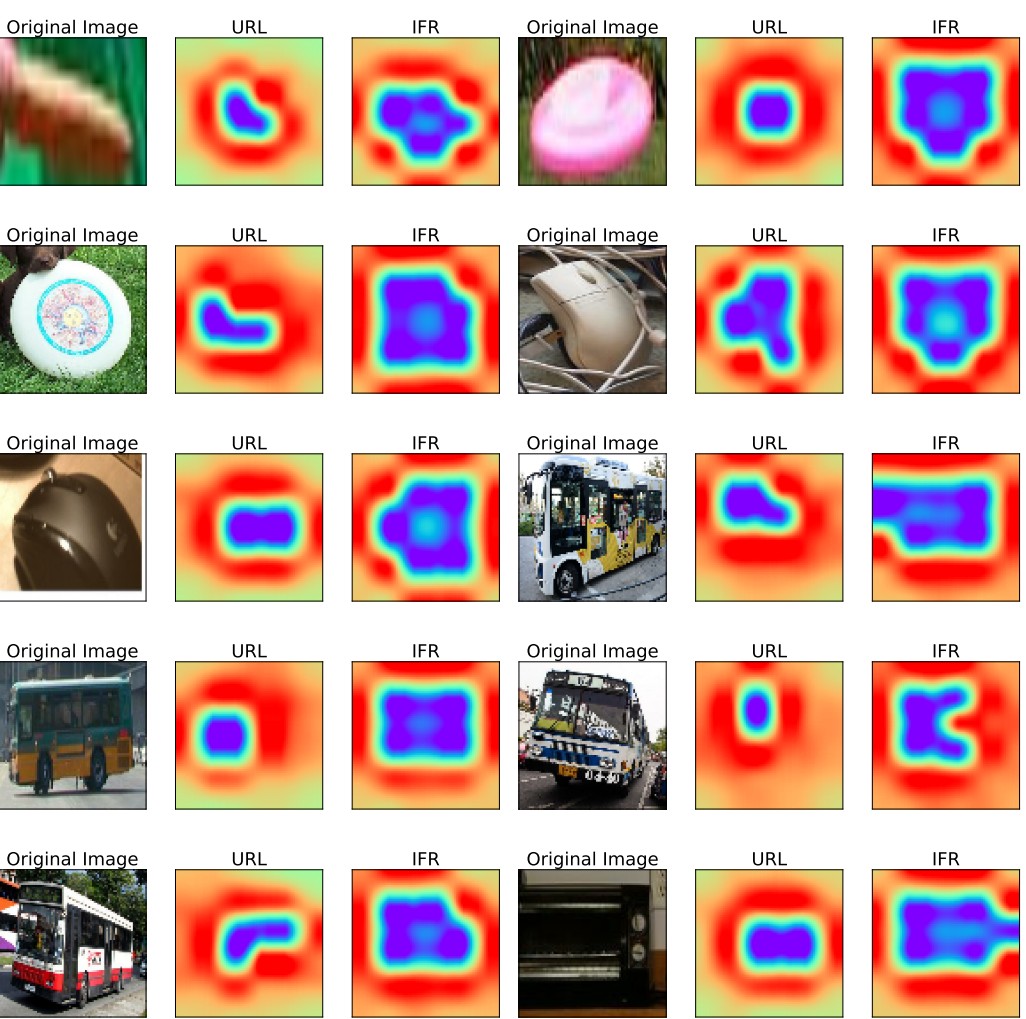

Figure 17: Comparions of features learned with URL and IFR on MSCOCO.

