# OpenReview forum: "Cross-domain Few-shot Classification via Invariant-content Feature Reconstruction"
_ICLR.cc/2024/Conference — Submitted to ICLR 2024_

### Official Review · Reviewer_kkCh · 2023-10-18

**Soundness:** 1 poor
**Presentation:** 2 fair
**Contribution:** 2 fair
**Rating:** 3
**Confidence:** 3

**Summary:**

This paper intends to combine high-level features and fine-grained invariant-content features to improve the performance of cross-domain few-shot classification. Specifically, the author proposes to extract invariant-content features via a single attention head and fuse the extracted invariant-content features and high-level features via the scaled dot-product attention mechanism of the Transformer. The proposed methodology recovers the key content features of the target class, which did not work well in existing meta-dataset works. Through extensive experiments, it is shown that the proposed method outperforms the baseline under various conditions.

**Strengths:**

1) This paper proposed a simple yet effective feature reconstruction method that significantly improves the performance.
2) This paper presents a theoretical analysis of the proposed attention modules.

**Weaknesses:**

1) The single attention head and scaled dot production attention mechanism proposed in this paper are not novel concepts and are so simple that the technical contribution is insufficient to acknowledge the quality.
2) The presented Theorem is just borrowed from the existing works.
3) It is not easy to know exactly what “high-level features” and “invariant-content features” denote. It is not clearly explained in the manuscript, and from the caption in Figure 2, one can only guess that the output features of the backbone are high-level features, and the output features of the attention head are invariant-content features.
3) Comparison with SoTA methods is insufficient. The baseline (URL, published in 2020) of the toy example in Figure 1 is outdated, and the experiment also needs to be compared with TSA[1] and TriM[2], which are currently recording higher performance than URL on the leaderboard.
4) Motivation is poor. The motivation of this paper is that prior works cannot capture representative key features, and the only evidence to support this is a comparison of the figure's activation map. This comparison alone is not sufficient to point out the shortcomings of prior works, and since it is a qualitative comparison, it is not accurate. As a result, the significant performance gains in Tables 1 and 2 are not credible, and even if it is true, the performance improvement cannot be sufficiently explained in the paper, so the contribution is greatly limited.
5) Citations are inconsistent. Some citations are contained within the parenthesis, while others are not.
There are redundant contents that are not very important. For example, Figure (4) is simple enough that it does not need to be shown as a figure, and there is no need to provide Theorem 2 since it is not proposed in this paper.


[1] Li, Wei-Hong, Xialei Liu, and Hakan Bilen. "Cross-domain few-shot learning with task-specific adapters." Proceedings of the IEEE/CVF Conference on Computer Vision and Pattern Recognition. 2022.
[2] Liu, Yanbin, et al. "A multi-mode modulator for multi-domain few-shot classification." Proceedings of the IEEE/CVF International Conference on Computer Vision. 2021.

**Questions:**

1) In the second paragraph on page 2, it is said that the proposed method considers both informativeness and discriminativeness at the same time. However, it is not easy to tell the difference between the two concepts through the current introduction. How exactly are they different, and how does the proposed method use these two concepts at the same time?
2) It is questionable whether the feature reconstruction mentioned in this paper is really reconstruction. To me, it just looks like performing a scaled dot product, and considering that the α value in Equation (2) is about 1e-4, I'm not sure if it has much significance.
3) Page 7 says that query heads have Lipschitz continuous property and the author’s comment follows as: “we can reliably leverage IFR to find good representations…”. But isn't this explanation too insincere and uninformative? Although it is known that Lipschitz continuity improves robustness against perturbation, but more detailed analysis is needed to be provided.

**Details Of Ethics Concerns:**

There is no ethics concern.

---

> ### Author Response · Authors · 2023-11-19
> **Rebuttal from authors to reviewer kkCh (1/3)**
>
> Thank you for your efforts in reviewing our work. We appreciate for your comments. Now, we post our responses for your concerns in the following.
>
> > __Weakness 1.__ The single attention head and scaled dot production attention mechanism proposed in this paper are not novel concepts and are so simple that the technical contribution is insufficient to acknowledge the quality.
>
> Per your question, we guess that there might be some understandings here.
>
> The core idea of our paper is __reconstructing fine-grained invariant-content features for few-shot classification__ instead of proposing a single attention head. __In fact, these basic and general modules, such as attention head and FiLM, have been widely used to realize different ideas in previous works, such as Tri-M and CrossTransformer etc.__.
>
> As mentioned in our paper, based on the assumptions: __(i) the two fundamental elements of an image are _content_ and _style_, (ii) content and style are independent, i.e. style changes are content-preserving__, a feasible way of reconstructing invariant-content features is __firstly locating the positions of invariant content features__ and __then retrieving these features to reconstruct invariant-content features__.
>
> Based on such idea, we choose to use a single attention head since __(i) the process of calculating similarity matrix is equivalent to measuring the similarities between each query and all keys to locate the most similar features in pixel-level__ and __(ii) the process of multiplying the similarity matrix to values is equavilent to retrieving features that are invariant to style modifications in the way of weighted sum__.
>
> > __Weakness 2.__ The presented Theorem is just borrowed from the existing works.
>
> Per your question, the main goal of the theorem is to __provide a direct and intuitive explanation of our concern that whether applying complex attention module (compared with linear transformation head) results in extremely large distance (e.g., infinity) between two original samples that share similar content features__ based on the existing theoretical results.
>
> According to the theorem proved in Vuckovic et al., 2021, we find that the distance between the features of two samples obtained from the attention module is upper boundedby the distance between the two original samples with a uniform Lipshitz constant. Thus, given a well pre-trained backbone (e.g., the pre-trained backbone proposed in URL or SUR), the distances extracted from the attention module will not be extremely large if the features themselves are close.
>
> > __Weakness 3.__ It is not easy to know exactly what “high-level features” and “invariant-content features” denote. It is not clearly explained in the manuscript, and from the caption in Figure 2, one can only guess that the output features of the backbone are high-level features, and the output features of the attention head are invariant-content features.
>
> Per your concern, for the "high-level features", a common sense in representation learning is that the output of a backbone is usually treated as high-level features that can depict the general semantic information of the input.
>
> For the "invariant-content features", we have described them in both introduction section and section 3.1. The description of "invariant-content features" is based on two assumptions proposed in ReLIC. To be brief, the "invariant-content features" are those discriminative features which depict the fundamental information of each image and are robust to style changes.
>
> > __Weakness 4.__ Comparison with SoTA methods is insufficient. The baseline (URL, published in 2020) of the toy example in Figure 1 is outdated, and the experiment also needs to be compared with TSA and TriM, which are currently recording higher performance than URL on the leaderboard.
>
> Per your question, in our method, we mainly compare IFR with those methods which train the entire pipeline from scratch or fine-tune a head on top of the pre-trained backbone. Thus, we did not take TSA into consideration. Thank you for your comments, we will further apply our IFR method to TSA to evaluate the performance TSA+IFR. As for TriM, we have compared IFR and TriM in experiment section of our paper(Table 1).

---

> ### Author Response · Authors · 2023-11-19
> **Rebuttal from authors to reviewer kkCh (2/3)**
>
> > __Weakness 5.__ Motivation is poor. The motivation of this paper is that prior works cannot capture representative key features, and the only evidence to support this is a comparison of the figure's activation map. This comparison alone is not sufficient to point out the shortcomings of prior works, and since it is a qualitative comparison, it is not accurate. As a result, the significant performance gains in Tables 1 and 2 are not credible, and even if it is true, the performance improvement cannot be sufficiently explained in the paper, so the contribution is greatly limited.
>
> Per your concern regarding our motivation derived from illustration figure, a fact that we have to agree is that many works in deep learning areas arise from intuitions that are not well backed up by empirical evidence. In our paper, we have provided as much evidence as possible to illustrate our motivations and effectiveness of our proposed method.
>
> Although Fig.1 seems insufficient to point out the shortcomings of prior works from your perspective, __it demonstrates that such phenomenon indeed exists__. Thus, __we utilize these figures to intuitively illustrate our main idea to readers__. Moreover, __as we mentioned in section 3.4, from the angle of model, since the previous work URL merely fine-tunes a linear transformation head on top of the fronzen pre-trained model, it is obvious that the function space in URL is less complexity__. We think both Fig.1 and the less complexity of function space of URL can demonsrate our motivation to some extent.
>
> As for the empirical results, we would like to note that __(i) our proposed method achieves better performance on unseen domains with obvious gaps in both Table 1 and 2; (ii) IFR also achieves better generalization performance on meta-dataset with other pre-trained models, and the results are reported in Table 12, Fig.5 and Fig.8__. According to these results, we do not think that the performance gains are not credible.
>
> > __Weakness 6.__ Citations are inconsistent. Some citations are contained within the parenthesis, while others are not. There are redundant contents that are not very important. For example, Figure (4) is simple enough that it does not need to be shown as a figure, and there is no need to provide Theorem 2 since it is not proposed in this paper.
>
> __[About incosistent citation.]__ Thank you for your comments, we will modifiy this in our future versions.
>
> __[About redundant content.]__ From our perspective, we do not completely agree that Theorem 2 is redundant content. The main goal of the theorem is to __provide a direct and intuitive explanation of our concern that whether applying complex attention module (compared with linear transformation head) results in extremely large distance (e.g., infinity) between two original samples that share similar content features__ based on the existing theoretical results.

---

> ### Author Response · Authors · 2023-11-20
> **Rebuttal from authors to reviewer kkCh (3/3)**
>
> > __Q1.__ In the second paragraph on page 2, it is said that the proposed method considers both informativeness and discriminativeness at the same time. However, it is not easy to tell the difference between the two concepts through the current introduction. How exactly are they different, and how does the proposed method use these two concepts at the same time?
>
> Per your question, an intuition about fine-grained featrues that are extracted in pixel-level is they contain more information. Besides, an implicit assumptions, which is widely adopted in learning representation for classification, is the class-specific representations are general among samples within the same class while discriminative enough between different classes. Since the invariant-content features are extracted in pixel-level and aims at learning robust features that depict the fundamental information of images, they are informative and discriminative.
>
> > __Q2.__ It is questionable whether the feature reconstruction mentioned in this paper is really reconstruction. To me, it just looks like performing a scaled dot product, and considering that the α value in Equation (2) is about 1e-4, I'm not sure if it has much significance.
>
> Per your question reagarding feature reconstruction, we have provided many visualization results in our paper to illustrate that IFR helps learn invariant-content features.
>
> Per your question regarding the hyperparameter $\alpha$, fine-tuning model with slight modification for transferability of prior knowledge is widely adopted in deep learning topics. For example, in TSA that you mentioned above, authors perform feature fusion of features respectively learned from frozen backbone modules and adapters with the coefficient 0.0001.
>
> Besides, since invariant-content features are fine-grained, if too large $\alpha$ is applied, the generalization performance will be negatively affected. You can observe such phenomenon in Section 4.2 in our paper.
>
> > __Q3.__ Page 7 says that query heads have Lipschitz continuous property and the author’s comment follows as: “we can reliably leverage IFR to find good representations…”. But isn't this explanation too insincere and uninformative? Although it is known that Lipschitz continuity improves robustness against perturbation, but more detailed analysis is needed to be provided.
>
> Per your question, the main goal of the theorem is to __provide a direct and intuitive explanation of our concern that whether applying complex attention module (compared with linear transformation head) results in extremely large distance (e.g., infinity) between two original samples that share similar content features__ based on the existing theoretical results.
>
> According to the theorem proved in Vuckovic et al., 2021, we find that the distance between the features of two samples obtained from the attention module is upper boundedby the distance between the two original samples with a uniform Lipshitz constant. Thus, given a well pre-trained backbone (e.g., the pre-trained backbone proposed in URL or SUR), the distances extracted from the attention module will not be extremely large if the features themselves are close.
>
> To be specific, a key step of IFR is measure the similarities among samples in pixel-level. Thus, by taking adavantage of the single attention head, we can guarantee that the original similar pixels, which depict the invariant-content features, are still close to each other so that we can locate the positions of the invariant-content features.

---

> ### Comment · Reviewer_kkCh · 2023-11-22
> **Post rebuttal**
>
> Dear authors,
>
> My biggest concerns were (i) limited technical novelty, (ii) lack of thoroughness of theoretical analysis, (iii) outdated baselines, and (iv) insufficient validity and representation of motivation. I’m afraid that my concerns are still not resolved by the author's answer. Therefore, I still keep my score as reject (3) and the reasons are as follows.
>
> ### **Limited technical novelty**
> The operating mechanism of IFR is composed of (i) locating content-invariant features, and (ii) feature reconstruction. However, in my view, (i,ii) is simply a separation of dot-product attention widely used in Transformers. Proposed pixel-wise dot-product attention appears to be a slight extension, and even this is very similar to crossTransformer's spatial-aware (pixel-wise) dot product. The details of IFR may be slightly different from prior works, however, considering that transformer-related methods have already been extensively studied in previous meta-dataset literatures, technological novelty cannot be acknowledged enough to give acceptance.
>
> ### **Lack of thoroughness of theoretical analysis**
> I understand that Theorem 2 in the manuscript did not directly borrow from Vuckovic et al.'s Theorem 14. However, I don't think Theorem 2 can be easily extended from prior work, and there is no proof of how to derive Theorem 2 in the manuscript or Appendix.
> Additionally, the fact that the authors didn’t mention the assumption of the cited theorem 14 degrades the completeness of the paper. From my perspective, contrary to the author's comment (“given the same mild assumptions…”), the assumption of theorem 14 does not seem to be a mild condition that can be easily applied to the deep neural network and transformer used in IFR.
> And even if Theorem 2 is valid, if the distance between background content is close, the attention value becomes similar. Therefore, IFR will generate a high correlation for background pairs between original images and augmented images. Therefore, I cannot trust the claim “IFR can find good representations of content-invariant features”.
>
> ### **Outdated baselines**
> I still think most baselines are outdated. All baselines except 2LM were published in 2020-2021, and there are no experimental results of 2LM in Table 2. I thought the author would add new experimental results during the discussion period, but it is disappointing that nothing has been added.
>
> ### **Insufficient validity and unreliable representation of motivation**
> The first thing I felt when I saw Figure 1 was that (i) only URL is used as the baseline and (ii) visualization of the features map is not a reliable observation to compare. Plus, It is not even well explained what this visualization represents. (e.g., is this correlation, the power of the value of the feature, or class activation map (CAM)?). And looking at the various figures in Figure 12, it seems that the outputs of URL and IFR are not well normalized. In most cases, similar results are expected to come out when the two outputs are normalized.
> As stated in the weakness 5 I mentioned, I still think the logic behind the motivation is not persuasive. During the discussion period, I expected the authors to at least quantitatively show that (i) the correlation between content areas (between the original and the augmented) is higher than that of background areas, or (ii) more comparison with other baselines. I was disappointed by the author's insincerity, and the current visualization still does not logically support the motivation of the paper.
>
> ### **Additional concern on hyperparameter $\alpha$ and TSA**
> In TSA, I couldn’t find that 0.0001 was used as the fusion weight but TSA instead optimized the coefficient parameter. (There is a mention in TSA's supplementary material that the residual adapter's scaler was initialized to 1e-4, but this value is not the final value.) Additionally, TSA is a concept of adding a residual adapter to all layers, so it is on the same line as IFR's feature fusion. There is no comparison.

---

### Official Review · Reviewer_LDzc · 2023-10-30

**Soundness:** 3 good
**Presentation:** 3 good
**Contribution:** 1 poor
**Rating:** 3
**Confidence:** 4

**Summary:**

This paper proposes an invariant-content feature reconstruction (IFR) method, which combines high-level semantic features with fine-grained invariant-content features for cross-domain few-shot classification.
The high-level semantic features are extracted from the original images by the backbone, and the invariant-content features are reconstructed from the augmented images by the transformer attention head.
In a word, IFR performs cross-attention between the original images and their augmented images.
The experimental results on the Meta-Dataset benchmark show the effectiveness of IFR in improving generalization performance on unseen domains.

**Strengths:**

1. The motivation and presentation are clear.
2. The experimental results are good: The experimental results on the Meta-Dataset benchmark show the effectiveness of IFR in improving generalization performance on unseen domains.

**Weaknesses:**

1. The novelty of the proposed methodology is limited. The proposed IFR method only performs cross-attention between the original images and their augmented images. The transformer-based cross-attention has been widely used.

**Questions:**

1. The proposed IFR method only performs cross-attention between the original images and their augmented images. What is the limitation of cross-attention in obtaining the fine-grained-content features? Can authors make an improvement to the cross-attention structure?

---

> ### Author Response · Authors · 2023-11-19
> **Rebuttal from authors to reviewer LDzc**
>
> > __Weakness 1.__ The novelty of the proposed methodology is limited. The proposed IFR method only performs cross-attention between the original images and their augmented images. The transformer-based cross-attention has been widely used.
> > __Q1.__ The proposed IFR method only performs cross-attention between the original images and their augmented images. What is the limitation of cross-attention in obtaining the fine-grained-content features? Can authors make an improvement to the cross-attention structure?
>
> Per your concern regarding the comparison between single attention head and cross-attention module, we guess that there might some misunderstandings here.
>
> The key motivation in our paper is __reconstructing fine-grained invariant-content features for few-shot classification tasks__ instead of just imply applying attention module.
>
> As mentioned in our paper, based on the assumptions: __(i) the two fundamental elements of an image are _content_ and _style_, (ii) content and style are independent, i.e. style changes are content-preserving__, a feasible way of reconstructing invariant-content features is __firstly locating the positions of invariant content features__ and __then retrieving these features to reconstruct invariant-content features__.
>
> Based on such idea, we choose to use a single attention head since __(i) the process of calculating similarity matrix is equivalent to measuring the similarities between each query and all keys to locate the most similar features in pixel-level__ and __(ii) the process of multiplying the similarity matrix to values is equavilent to retrieving features that are invariant to style modifications in the way of weighted sum__.

---

### Official Review · Reviewer_LoxD · 2023-11-01

**Soundness:** 3 good
**Presentation:** 3 good
**Contribution:** 3 good
**Rating:** 6
**Confidence:** 4

**Summary:**

The paper addresses the challenge of cross-domain few-shot classification (CFC), where the objective is to perform classification tasks in previously unseen domains with limited labeled data. The authors propose a novel approach named Invariant-Content Feature Reconstruction (IFR), which aims to simultaneously consider high-level semantic features and fine-grained invariant-content features for unseen domains. The invariant-content features are extracted by retrieving features that are invariant to style modifications from a set of content-preserving augmented data at the pixel level using an attention module. The paper includes extensive experiments on the Meta-Dataset benchmark, demonstrating that IFR achieves superior generalization performance on unseen domains and improves average accuracy significantly under two different CFC experimental settings.

**Strengths:**

- Novel Approach: The paper introduces a unique method, IFR, which addresses the limitations of existing approaches by considering both high-level semantic features and fine-grained invariant-content features. This dual consideration is innovative and addresses a critical gap in cross-domain few-shot classification.

- Extensive Experiments: The authors have conducted comprehensive experiments on the Meta-Dataset benchmark, providing a robust evaluation of their proposed method. This adds credibility to their claims and demonstrates the practical applicability of their approach.

- Clear Problem Statement: The paper clearly articulates the challenges in cross-domain few-shot classification and provides a compelling argument for why existing methods are insufficient, setting a strong foundation for their proposed solution.

**Weaknesses:**

- Limited Explanation of Methodology: While the paper provides a high-level overview of the IFR approach, it could benefit from a more detailed explanation of the methodology, including the attention module and how it specifically contributes to invariant-content feature extraction.

- Lack of Comparative Analysis: The paper presents experimental results demonstrating the effectiveness of IFR, but it lacks a thorough comparative analysis with existing methods, discussing in detail why IFR outperforms them. In addition, since this proposed work resembles [1][2], why not compare with these two methods in the Experiments Section?

- Potential for Overfitting: Given that the approach focuses on fine-grained features, there might be a risk of overfitting, especially when dealing with extremely limited data in few-shot scenarios. The paper does not address this potential issue or provide strategies to mitigate it.

[1] Memrein: Rein the domainshift for cross-domain few-shot learning. IJCAI 2020
[2] Cross-domain few-shot classification via adversarial task augmentation. IJCAI 2021

**Questions:**

1. Can you provide a more detailed explanation of the attention module used in IFR and how it specifically contributes to the extraction of invariant-content features?

2. How does IFR compare to existing methods in terms of computational efficiency and scalability, especially when applied to large-scale datasets?

3. Given the focus on fine-grained features, how does IFR mitigate the risk of overfitting in few-shot scenarios? Are there any specific strategies or mechanisms in place to prevent this?

4. How well does IFR generalize to domains that are significantly different from those in the Meta-Dataset benchmark? Have there been any experiments conducted in this regard?

---

> ### Author Response · Authors · 2023-11-19
> **Rebuttal from authors to reviewer LoxD (1/3)**
>
> Thank you for your effort in reviewing our work. We appreciate your valuable suggestions and comments. Now, we post our responses for your concerns in the following.
> > __Weakness 1 & Q1.__ Limited Explanation of Methodology: While the paper provides a high-level overview of the IFR approach, it could benefit from a more detailed explanation of the methodology, including the attention module and how it specifically contributes to invariant-content feature extraction. / Can you provide a more detailed explanation of the attention module used in IFR and how it specifically contributes to the extraction of invariant-content features?
>
> __[About attention module.]__ Per your question about attention module, we have explained it in our experimental setting part. To be specific, the attention module adopted in our paper is composed of 3 linear layers with the same dimension $512 \times 512$. Following URL, in order to take advantage of the features extracted from the frozen pre-trained backbone, all linear layers in the attention module are initialized as identity matrix at the beginning of each learning episode.
>
> <font color=blue>__[About function of attention module.]__</font> Regarding to problem of how the attention module specifically contributes to the extraction of invariant-content features, __we have made a detailed explanation in Section 3__. __The main contribution is that the attention module fits the procedure of invariant-content features quite well__.
>
> First of all, an important assumption of our method is: __(i) the two fundamental elements of an image are _content_ and _style_, (ii) content and style are independent, i.e. style changes are content-preserving__. According to these assumptions, content features that depict the fundamental information of an image are invariant to style modifications.
>
> Generally, the reconstruction of invariant-content features mainly include two steps: __locating the positions of invariant content features__ and __retrieving these features__. In this paper, the two steps are realized with a single attention module.
>
> Specifically, we first locate the positions of invariant-content features by measuring the similarities between original and augmented image representations in pixel level. As illustrated in Fig. 3(a), given a set of queries embedding and keys embeddings, $\boldsymbol{Q}\in\mathbb{R}^{w\times h\times d}$ (transformed from original images) and $\boldsymbol{K}\in\mathbb{R}^{w\times h\times d}$ (transformed from augmented images), a similarity matrix $\boldsymbol{M}_{\rm sim}\in\mathbb{R}^{wh\times wh}$ is measured to describe the similarity between arbitrary two pixels respectively from queries and keys embedding. As the aforementioned assumptions, content features are invulnerable to the changes of style. Thus, two pixels that depict the same content information ought to be highly similar to each other and the similarity score will be relatively larger than those depict trivial style information. Thus, the positions of pixels that depict the same invariant-content features in augmented representations are determined for each pixel of original representations.
> Then, with the similarity matrix, the invariant-content features are reconstructed by retrieving pixels with corresponding similarity scores (as shown in Fig. 3(a)). The reconstruction process is equivalent to a recombination of pixel vectors where the invariant-content features assigned with larger similarity scores are highlighted (as shown in Fig. 3(b).

---

> ### Author Response · Authors · 2023-11-19
> **Rebuttal from authors to reviewer LoxD (2/3)**
>
> > __Weakness 2.__ Lack of Comparative Analysis: The paper presents experimental results demonstrating the effectiveness of IFR, but it lacks a thorough comparative analysis with existing methods, discussing in detail why IFR outperforms them. In addition, since this proposed work resembles [1][2], why not compare with these two methods in the Experiments Section?
>
>
> __[About comparative analysis.]__ Thank you for your constructive suggestion. As we mentioned in our paper, __the main discrepancy between IFR and previous URL is that IFR further considers capturing more discriminative and informative fine-grained invariant-content features that are robust to style modifications and depict the fundamental information of the image__. Meanwhile, __compared with IFR which merely fine-tunes a simple linear transformation head on top of a pre-trained backbone, IFR, which trains a more complicated module, owns much more complex function space to fit the representations__. Thus, IFR obtains better results.
>
> In our paper, both quantative results in tables and visualization results of learned representations demonstrate our statement. Besides, we also have conducted several ablation studies to explore the effectiveness of each part of our proposed modules.
>
> __[About comparison to the mentioned references.]__ Per your concern regarding the comparison between IFR and the given two references, in fact, we have provided discussions in Appendix A (Paragraph 2, Page 15).
>
> We agree that the insights behind our proposed IFR are similar to those in [1, 2]. All of these methods aims to __drop the unimportant style informtion like colors and preserve the robust and invariant features with good discrimination ability__. __However, there are some differences in the ways to reach the goal__. For example, MemREIN proposed in [1] reduces the cross-domain descrepancies(colors etc.) by performing instance normalization and then uses a memorized restitution to preserve the discrimination ability of the refined features, and the adversarial task augmentation proposed in [2] takes the idea to learn robust features by generating the inductive bias-adaptive 'challenging' tasks. But in our IFR, we use a simpler way. We weaken the style information and highlight the invariant-content features by extracting the invariant information between the original data and their augmented counterparts with a single-head attention module. For the method proposed in [3], we think it is quite different from [1, 2] and our proposed IFR. Different from [1, 2] and our IFR that tries to extract more robust and useful features from the original features, the method proposed in [3] introduces additional hyperparameters to learn feature transformation. The proposed feature wise transformation resembles the FiLM proposed in [4].
>
> Per your question regarding comparing two methods in experiment section, since we __mainly focus on cross-domain few-shot classification tasks with vary-way vary-shot settings__ proposed in meta-dataset[5], we did not take these two methods taken into consideration.

---

> ### Author Response · Authors · 2023-11-19
> **Rebuttal from authors to reviewer LoxD (3/3)**
>
> > __Weakness 3 & Q3.__ Potential for Overfitting: Given that the approach focuses on fine-grained features, there might be a risk of overfitting, especially when dealing with extremely limited data in few-shot scenarios. The paper does not address this potential issue or provide strategies to mitigate it.
>
> Thank you for pointing out this, we also notice that there exists overfitting phenomenon and such phenomenon further negatively affect the performance of our method. __The main reason that we did not apply any technique to mitigate such phenomenon is that few work, as far as we know, has applied strategies for mitigating overfitting during meta-test phase__. A feasible way for such case is to consider modifying the adaptation loss to supress data overfitting. We will think about this problem in future version of our paper.
>
>
> > __Q2.__ How does IFR compare to existing methods in terms of computational efficiency and scalability, especially when applied to large-scale datasets?
>
> __[About computational efficiency.]__ In order to answer your question, we roughly compare the number of parameters used in pre-training based learning frameworks, including FLUTE, URT, URL and our proposed IFR. We list the results in the following.
>
> | **Method** | **URL** | **IFR** | **FLUTE** | **URT** |
> | -------- | -------- | -------- | -------- | -------- |
> | **Num Params.** | $8\times 2^{15}$ | $\sim 32\times 2^{15}$ | $\sim 45\times 2^{15}$ | $\sim 416\times 2^{15}$ |
>
> As we can observe from the table, __the number of parameters of IFR is about 4 times over URL which only simply trains a linear transformation head on top of the frozen pre-trained backbone__. Meanwhile, __in IFR, the size of features extracted from the frozen pre-trained backbone is `6 x 6` instead of `1 x 1` as in URL.__ Thus, it is reasonable that IFR takes more time. Under the same hardware settings, URL takes about 1.5 hours for adaptation of all 13 datasets while IFR takes about 7 hours.
>
> __[About stability.]__ Per your concern, all results reported in tables in our paper are the averaged performance of 10 random seeds. Thus, we think that the stability is fine.
>
> __[About scalability.]__ Since IFR can be scaled to any few-shot classification tasks, from our perspective, the scalability is fine. For example, the ImageNet in meta-dataset is a kind of large-scale dataset.
>
> > __Q4.__ How well does IFR generalize to domains that are significantly different from those in the Meta-Dataset benchmark? Have there been any experiments conducted in this regard?
>
> This is an interesting question. From our perspective, __an implicit assumption shared in current cross-domain few-shot classification works is that the downstream tasks evaluated during meta-test phase ought to be similar in high level to those which have been observed by the model__. Thus, if the target dataset has a significantly large distribution gap with those datasets in meta-dataset, it is reasonable that the performance drops since the pre-trained model may fail to capture useful features for further adaptation.
>
> For example, under "Train on ImageNet only" settings, the pre-trained model has only observed data from ImageNet dataset. Thus, datasets, such as Omniglot and MNIST, can be approximately treated as datasets that are significantly different from datasets in meta-dataset. Then, we can easily observe that the performance drops significantly compared with that under "Train on all datasets" settings. We think such observation can help address your concern.
>
> #### Reference
>
> [1] Memrein: Rein the domainshift for cross-domain few-shot learning. IJCAI 2020
>
> [2] Cross-domain few-shot classification via adversarial task augmentation. IJCAI 2021
>
> [3] Tseng et al. Cross-domain few-shot classification via learned feature-wise transformation. ICLR 2020
>
> [4] Perez et al. Film: Visual reasoning with a general conditioning layer. AAAI 2018

---

### Meta-Review · Area_Chair_r7MS · 2023-12-05

**Metareview:**

Dear Authors,

Thank you for submitting the draft, on an important problem. 2 out of three reviewers have assigned rank 3 (reject, not good enough) and one has assigned rank 6 (marginally above the acceptance threshold). After analyzing the reviews, and rebuttal itself, we agree that the draft is currently not ready for publication. One of the objections by the reviewer is that only one recent result is there in the comparison. The readability of the draft could be improved.

We suggest authors update the draft by taking into consideration the concerns of the reviewers.

regards
AC

**Justification For Why Not Higher Score:**

Low scores from reviewers.

**Justification For Why Not Lower Score:**

N/A

---

### Decision · Program_Chairs · 2024-01-16

Reject